# Study of Carbon Reduction and Marketing Decisions with the Envisioning of a Favorable Event under Cap-and-Trade Regulation

**DOI:** 10.3390/ijerph20054644

**Published:** 2023-03-06

**Authors:** Weihao Wang, Deqing Ma, Jinsong Hu

**Affiliations:** School of Business, Qingdao University, Qingdao 266071, China

**Keywords:** carbon reduction, marketing, cap-and-trade regulation, random stopping time optimal control, differential game, supply chain management

## Abstract

To achieve SDGs (sustainable development goals) and carbon neutrality goals, the Chinese government have been adopting the cap-and-trade regulation to curb carbon emissions. With this background, members in the supply chain should properly arrange their carbon reduction and marketing decisions to acquire optimal profits, especially when the favorable event may happen, which tends to elevate goodwill and the market demand. However, the event may not be of their benefit when the cap-and-trade regulation is conducted, since the increase in market demand is always associated with an increase in carbon emissions. Hence, questions arise about how the members adjust their carbon reduction and marketing decisions while envisioning the favorable event under the cap-and-trade regulation. Given the fact that the event occurs randomly during the planning period, we use the Markov random process to depict the event and use differential game methodology to dynamically study this issue. After solving and analyzing the model, we acquire the following conclusions: (1) the occurrence of the favorable event splits the whole planning period into two regimes and the supply chain members should make optimal decisions in each regime to maximize the overall profits. (2) The potential favorable event will elevate the marketing and carbon reduction efforts, as well as the goodwill level before the event. (3) If the unit emissions value is relatively low, the favorable event will help to decrease the emissions quantity. However, if the unit emissions value is relatively large, then the favorable event will help to increase the emissions quantity.

## 1. Introduction

In 2015, the United Nation suggested 17 Sustainable Development Goals (SDGs) to offer a guide for sustainable development and tackle the issues during economic growth. Some of the SDGs are closely related to carbon emission reduction [1,2,3]. For example, Goal 12 requires sustainable production patterns and Goal 13 advocates to take actions to combat climate change [4]. China, the largest developing economy and therefore the biggest carbon emitter in the world, is faced with the challenge of balancing development and carbon reduction, since economic growth and environment protection are both important goals according to the 14th Five-Year Plan [5]. To achieve the SDGs and the carbon neutrality goal, the Chinese government has been taking measures to solve this problem, among which the carbon tax and cap-and-trade policies are the two major methods in practice. However, compared to the carbon tax policy, the cap-and-trade regulations are more effective in curbing carbon emissions [6,7,8]. Since 2011, the Chinese government has piloted this regulation in many provinces and cities [9,10], and officially operated the trading market in the power industry in 2021 [11].

Under the cap-and-trade regulation, each company will be assigned a cap on the amount of carbon emissions that it can produce, and be allowed to buy or sell the allowances in the carbon trading market. If the emission quantity of the firm exceeds the emission cap, it should buy more allowances from another firm with extra allowance in the trading market; otherwise, it can keep the spare allowances to cover its future needs or sell them to another company who is short of allowances in the market to acquire more revenue [12,13]. Nowadays, the cap-and-trade has been roundly carried out. For example, according to the Tesla First Quarter Report in 2022, the leading electric vehicle automaker gained USD 679 million in the first quarter by selling the carbon allowance to other automakers in the carbon trading market [14].

Essentially, the cap-and-trade regulation requires companies to balance the investment in production and the investment in carbon reduction [15]. Hence, under this regulation, each company should integrate the product market and the carbon trading market to maximize the profit, indicating that it is no longer profitable for companies to manufacture at the sacrifice of heavy carbon emission, and companies are forced to change their production process and produce green products [16]. 

The study of the cap-and-trade regulation has drawn much attention and many scholars have contributed much to this area [17,18,19,20,21,22,23]. However, based on the extant literatures, we focus on an interesting phenomenon. During the operation process, each company may encounter a favorable event that can elevate the goodwill level and therefore the market demand. For example, ERKE, a clothing brand in China, went viral on the internet for its donation to a flooded area in 2021 and triggered “wild consumption” of the brand on the internet. This kind of event is unexpected and can enhance the brand goodwill enormously and instantly. However, an increase in demand is always associated with an increase in carbon emissions, indicating that the company may suffer a loss or gain less revenue in the carbon trading market if he does not increase the carbon reduction effort. As a result, questions arise about how the members deal with an event that can elevate market demand and carbon emissions simultaneously, and how the members adjust their carbon reduction and marketing decisions with the change of probability and influence of such an event under the cap-and-trade regulation. To answer the questions above, a differential game model with consideration of a favorable event depicted by the Markov process is set up under the cap-and-trade regulation and the random stopping time optimal control theory is used to solve the problem.

The contribution of this research to extant research can be concluded as:(1)We base our research on the studies of cap-and-trade regulation for we refer to the extant literatures to construct the basic model of cap-and-trade regulation. Meanwhile, we further incorporate the favorable event into consideration and discuss the impacts of this event on the carbon reduction and marketing decisions under this regulation. Since the favorable event can elevate the goodwill and market demand, it will also enhance the carbon emission simultaneously. Hence, it is necessary to discuss the operational strategies of supply chain members when facing such an event. The contribution of this research is to offer suggestions to tackle the favorable event under the cap-and-trade and enrich the literatures in this stream.(2)Meanwhile, this research contributes to the literatures using the differential game model to study the sustainable supply chain. Extant research has studied in depth the carbon reduction and marketing decisions in a sustainable supply chain by using the differential game model. Based on the literatures in this stream, we set up a differential game model under the cap-and-trade regulation with consideration of the favorable event and study the carbon reduction and marketing decisions under the influence of the event.(3)This research also contributes to the study of the random stopping time optimal control problem. Extant literatures adopting this optimal control method, such as [24,25,26], mostly focus on the study of a crisis event instead of a favorable event. Actually, both crises and favorable events are likely to happen. Hence, this research bridges this gap by studying the favorable event in the sustainable supply chain under the cap-and-trade regulation.

The rest of this paper is organized as follows. Section 2 discusses the related literature. Section 3 formulates the model and describes the assumptions. Section 4 studies the scenario with no favorable event under the cap-and-trade regulation. Section 5 studies the scenarios with presence of a favorable event under the cap-and-trade regulation and derives the optimal results of both members before and after the event. Section 6 compares the optimal results under different circumstances. Section 7 numerically analyzes previous results and the impacts of the event on optimal results and goodwill. Section 8 concludes this paper.

## 2. Literature

This research is closely related to four streams of the literature: cap-and-trade regulation, the differential game model of sustainable supply chain, marketing in a sustainable supply chain, and random stopping time optimal control.

### 2.1. Cap-and-Trade Regulation

Cap-and-trade regulation is widely implemented and regarded as an important method to mitigate the pollution problem, hence intriguing many researchers to study this issue in sustainable supply chain management. The study of cap-and-trade concerns various aspects, among which investigation concerning the carbon reduction technology investment decisions under the cap-and-trade regulation is closest to this study. Hence, we mainly sort out the research in this stream. Concerning this stream, Xu et al. [27] studied the production and emission reduction decisions of a make-to-order supply chain consisting of a manufacturer and a retailer under the cap-and trade regulation and discuss several contracts to coordinate the supply chain. Dong et al. [28] studied the quantity order of the retailer and the sustainability investment of the manufacturer under cap-and-trade regulations in centralized and decentralized scenarios, respectively. They found that the sustainability investment efficiency has a significant impact on the optimal solutions. Yang et al. [29] studied the manufacturer’s channel selection and carbon reduction strategies with consideration of consumers’ low-carbon preference under the cap-and-trade regulation. They consider three models, which are single retail channel, single online channel and dual-channel models, respectively, and found that the channel conflict appears to have been mitigated under the cap-and-trade regulation. Xia et al. [30] studied the pricing and emission reduction decisions of a supply chain in a social environment under a cap-and-trade regulation with consideration of reciprocal preferences and discussed the impacts of this preference on optimal decisions and utilities. Li et al. [31] studied green technology and green marketing decisions under the cap-and-trade regulation with consideration of the government subsidy. Ding et al. [23] studied the manufacturer’s encroachment and carbon emission reduction decisions under cap-and-trade regulation with consideration of consumers’ low-carbon preference.

Inspired by extant literatures mentioned above, this paper also concentrates on the carbon reduction investment decision under the cap-and-trade regulation. However, this paper further incorporates the potential favorable event into consideration and discusses the impacts of the suddenly occurring event on the carbon reduction under the cap-and-trade regulation by using the differential game model.

### 2.2. Differential Game of Sustainable Supply Chain

Differential game is an effective and hence usually adopted method for studying the carbon reduction strategy in the sustainable supply chain. For example, Zhou and Ye [32] focused on a dual-channel low-carbon supply chain system and studied the joint carbon reduction strategies by utilizing the differential game model. They discussed the effects of the cooperative advertising contract and the cooperative advertising carbon reduction cost sharing contract on the decision-making and coordination of dual-channel supply chains. Xia et al. [33] studied the upstream–downstream joint carbon reduction strategies of a manufacturer producing low-carbon products and a retailer. They found that the carbon reduction per product unit is the highest when the supply chain is totally coordinated. Xia et al. [34] incorporated social preference in the study of the sustainable supply chain and analyzed the carbon reduction and promotion strategies of supply chain members by using differential game. He et al. [35] studied the contract design problem in a low-carbon service supply chain and derived the optimal carbon reduction and service decisions of the differential game model under different contract mechanisms. Wei and Wang [36] studied the impact of the regulation level on the carbon reduction decisions in a differential game model, taking the regulation of the government into consideration. Ma et al. [37] set up a differential game model of the sustainable supply chain considering technology and government intervention and studied the optimal carbon reduction and investment decision under different scenarios. Zu [38] studied the wholesale price contract and the consignment contract in a two-echelon supply chain consisting of a manufacturer and a retailer with consideration of the effect of emission reduction. He [39] established the differential game model to study the impact of bilateral participation strategy on the carbon reduction decision and associated performance of low-carbon supply chains. There are also studies solving the optimization problems under the cap-and-trade regulation by using the differential model [40,41,42].

Since the favorable event may happen at any instant during the planning period, it is appropriate to use this method to study this kind of random event. Hence, we set up a differential game model with consideration of the favorable event and discuss how the unexpected event may affect the supply chain members’ decisions under the cap-and-trade regulation.

### 2.3. Marketing in the Sustainable Supply Chain

Study of marketing decision is also an important stream in the management of the sustainable supply chain, which is closely relevant to this paper. Among the research concerning this stream of study, Basiri and Heydari [43] studied the green marketing decision and coordination strategy in a low-carbon supply chain system producing and selling two substitutable products. Hong and Guo [44] studied several cooperation mechanisms in a green supply chain and analyzed the effects of different mechanisms on the environmental performance of the supply chain and the retailer’s marketing decisions. Guo et al. [45] studied the effects of eco-labels on marketing decisions and the supply chain’s profit from both the profitability and environmental perspectives. They found that when the agency contract is used and consumer green awareness increases, the sale price declines rather than increasing. Wang and Song [46] explore pricing and marketing decisions in a dual-channel supply chain system with consideration of demand uncertainty. Li et al. [47] focused on a green product supply chain where market demand depends on the product greening level and the marketing effort. They found that enhancing the product greening level can benefit both supply chain members no matter whether the marketing effort is high or low. Wang et al. [40] investigated the carbon reduction and marketing effort in a green supply chain under the cap-and-trade regulation by using the differential game model. Liu et al. [48] explored the impact of cross-shareholdings on the carbon technology investment and green marketing decisions. Kou et al. [49] constructed Stackelberg game models with and without the retailer’s green marketing and obtained the equilibrium results by the reverse induction method, and explored the value of cooperative carbon reduction under the cap-and-trade regulation. Shi et al. [50] studied the green product development and green marketing strategies of a supply chain consisting of one manufacturer and one retailer to explore who is more suitable to implement these two strategies.

In line with extant literatures concerning the study of marketing effort in the sustainable supply chain, we study the dynamic marketing effort under the cap-and-trade regulation. Furthermore, we incorporate the impact of a potential favorable event into consideration and discuss the impact of this event on green marketing decisions.

### 2.4. Random Stopping Time Control Problem

Another stream of research relevant to this paper is the study of an optimal control problem with a random stopping time, in which the Markov process is introduced to characterize the occurrence of an uncertain event occurring suddenly. Many researchers have contributed to the study of this type of optimal control problem. For example, some scholars focus on the solution of this problem [51,52,53]. They deal with the stochastic optimal control problem where randomness is essentially concentrated in the stopping time terminating the process by applying maximum principle and dynamic programming techniques. The basic thought adopted is to turn the stochastic control problem into the deterministic problem; then, the study of this control problem is extended to the differential game model [54,55,56,57,58]. Since the Markov process is desirable to depict the randomly occurring event, the method is widely adopted in the study of an unexpected event and corresponding impacts, such as potential competitive entry [59,60,61,62], a product recall crisis [26,63,64], climate change or environmental disaster prevention [65,66,67], and the impact of the COVID-19 pandemic [68]. Different from the literatures mentioned, this paper focuses on a favorable event which can elevate the goodwill level almost overnight and explores the relationship between the event and the carbon reduction decisions under the cap-and-trade regulation. 

## 3. Assumptions and Notations

### 3.1. Basic Assumptions

To analyze the impacts of a favorable event upon the operational strategies in a low-carbon supply chain under the cap-and-trade mechanism, we construct a differential game model of a supply chain system consisting of a manufacturer and a retailer, in which the manufacturer produces one featured product and sells its product to the consumers through a retailer. The manufacturer invests in carbon reduction technology to reduce carbon emissions during the production process, which in fact decreases the unit cost of a product. At the same time, the retailer undertakes a green marketing effort to enhance goodwill closely relevant to demand.

**Assumption** **1.***The retailer decides on a green marketing effort which positively contributes to the accumulation of goodwill. Hence, by using the Arrow and Nerlove model, the dynamics of goodwill influenced by the green market effort is depicted by the following differential equation:*(1)$$\dot{G}\left(t\right)=\gamma A_{R}\left(t\right)-\delta G\left(t\right),G\left(0\right)=G_{0}$$*where*G(t)*is the stock of goodwill at time*t*with the initial level*$G_{0}>0$. $A_{R}\left(t\right)$*represents the green marketing effort determined by the retailer and*γ*denotes the marginal effectiveness of the marketing effort on the goodwill.*δ>0*is the decay rate of goodwill.*

**Assumption** **2.**
*The demand is positively associated with the level of goodwill. Meanwhile, the manufacturer’s carbon reduction technology can also enhance the market demand if the consumers are sensitive to the environment protection effort. Hence, influenced by goodwill and the carbon reduction effort, the dynamics of demand can be depicted by the following equation:*

(2)
D(t)=a+θG(t)

*where*

D(t)

*denotes the market demand at time*

t

*and*

a>0

*is basic demand.*

θ>0

*is the marginal contribution of goodwill to the demand.*

$I_{M}\left(t\right)$

*represents the manufacturer’s carbon reduction effort and*

β>0

*represents the consumers’ environmental preference, the higher degree of which indicates higher influence of manufacturer’s effort on demand.*


**Assumption** **3.***According to the literature [69,70], we assume that the manufacturer’s carbon reduction effort contributes to the reduction of the unit carbon emission of a product. Hence, the total carbon emitted at time*t*can be expressed by the following equation:*(3)$$E\left(t\right)=\left[\phi -\lambda I_{M}\left(t\right)\right]D\left(t\right)$$*where*E(t)*represents the carbon emission at time*t. ϕ>0*represents the original unit carbon emissions with the absence of carbon reduction technology.*λ>0*denotes the effectiveness of carbon reduction on the reduction of unit emissions. Meanwhile, according to the benchmarking rule [71,72,73], the unit carbon quota is allocated to the manufacturer, and we assume the baseline quota to be*$\phi_{B}$.

**Assumption** **4.**
*Consistent with assumptions in previous studies [32,33,34], we assume that the cost of the manufacturer’s carbon reduction effort and the retailer’s green marketing effort are related to their effort level, characteristic of diminishing returns. Hence, we take the quadratic form and assume the cost of the carbon reduction effort and green marketing effort to be:*

(4)
$$C_{M}\left(t\right)=\frac{k_{M}}{2}I_{M}^{2}\left(t\right),C_{R}\left(t\right)=\frac{k_{R}}{2}A_{R}^{2}\left(t\right)\;$$

*where*

$k_{M}$

*and*

$k_{R}$

*are positive constants representing the cost coefficients. We also assume*

$\rho_{M}$

*and*

$\rho_{R}$

*to be marginal profits of the manufacturer and the retailer, respectively.*


### 3.2. Regime Switch as a Result of Favorable Event

In this section, we incorporate the favorable event in our analysis. Since the favorable event occurs at a random time, we introduce the random occurrence process to depict this event.

**Assumption** **5.***Consistent with the common assumptions in previous literature related to the random stopping time optimal control problem [51,52,53], for simplicity of analysis, we assume that the favorable event happens once at time*t*, and the occurrence process is defined by the following conditional probability:*(5)$$\underset{\Delta t\to 0}{\lim}\frac{P\left\{t\leq T<t+\Delta t|T\geq t\right\}}{\Delta t}=\chi$$*where*χ>0*represents the occurrence rate and*T≥0*is the exact time at which the favorable event happens. Equation (5) indicates the probability of the event happening in the time interval*[t,t+Δt)*, given the fact that it has not taken place before, from which we can derive the probability density function*$f\left(t\right)=\chi e^{-\chi t}$*, the probability distribution function*$F\left(t\right)=1-e^{-\chi t}$*and the expectation*E(t)=1/χ*. Hence, the probability of the event happening before time*t*is*$P\left\{T\leq t\right\}=1-e^{-\chi t}$*, indicating that the event happens earlier and the expectation value*E(t)=1/χ* is smaller with the higher occurrence rate. Meanwhile, the favorable event divides the planning period into two regimes, and the goodwill will be enhanced due to the occurrence of the event. Hence, the relations of goodwill before and after the event can be characterized by the following equation:*(6)$$G\left(T^{+}\right)=\left(1+\varepsilon \right)G\left(T^{-}\right)$$*where*$G\left(T^{+}\right)$*represents the goodwill level right after the event and*$G\left(T^{-}\right)$*represents the goodwill level right before the event.*ε>0*is the expansion rate of the goodwill level. Since the favorable event can elevate the goodwill level, we use the equation above to depict the influence of the event on the goodwill, which is different from the studies of crisis which exert damage on the goodwill [24,25]. Hence, from the equation above, we can easily find that the goodwill will not be continuous due to the event, which is different from the usual assumptions in the previous literature. Meanwhile, it is also assumed that the event will not only enhance the goodwill level, but will also enhance the marginal effectiveness of the marketing effort on goodwill. Let*$\gamma_{1}$*and*$\gamma_{2}$*be the marginal effectiveness of marketing effort on goodwill, then we will have*$\gamma_{1}<\gamma_{2}$.

### 3.3. Objective Functionals of Supply Chain Members

When the occurrence rate χ=0, which means the favorable event will not happen, the goodwill is continuous in the whole planning period, and therefore the objective functionals of supply chain members can be expressed as: (7)$$J_{i}=\int_{0}^{\infty }e^{-rt}\pi_{i}\left(G,A_{R},I_{M}\right)dt$$
where i=M,R, representing the manufacturer and retailer, respectively. $J_{i}$ represents the profits of supply chain members during the planning period discounted to the initial time and r>0 denotes the discount rate. However, if the event is likely to happen, the supply chain members should calculate the optimal profits of two regimes to maximize the expected profit in the whole planning period. Hence, the objective functionals of supply chain members before and after the event can be expressed as: (8)$$J_{i1}=\int_{0}^{T}e^{-rt}\pi_{i1}\left(G_{1},A_{R1},I_{M1}\right)dt,\;J_{i2}=\int_{T}^{\infty }e^{-r\left(t-T\right)}\pi_{i2}\left(G_{2},A_{R2},I_{M2}\right)dt$$
where $J_{i1}$ represents the profits before the event, discounted to the initial time, and $J_{i2}$ represents the profits after the event, discounted to time T. Hence, the expected value of profits regarding random variable T can expressed as: (9)$$J_{i}=E\left[J_{i1}+e^{-rT}J_{i2}\right]$$

After calculating the expectations by using the probability density function and the probability distribution function in Section 3.2, we obtain the objective functionals of supply chain members with the envisioning of a favorable event as: (10)$$J_{i}=\int_{0}^{\infty }e^{-\left(r+\chi \right)t}\left[\pi_{i1}\left(G_{1},A_{R1},I_{M1}\right)+\chi J_{i2}\right]dt$$

In the next section, we will further solve and analyze the model to explore the influences of a favorable event upon the carbon reduction and marketing decisions, as well as the supply chain members’ optimal profits. In addition, time variable t will be omitted in the analysis below, where no ambiguity will be caused. All parameters and variables involved are listed in the following Table 1.

## 4. Base Model

In this section, we first explore the scenario without consideration of the favorable event (superscript N is used to denote this scenario), and the goodwill is continuous during the whole planning period. In this scenario, both supply chain members make decisions separately to maximize their own net profit, in which the retailer decides on the marketing effort and the manufacturer decides on the carbon reduction effort. We use $\phi_{M}$ to represent the baseline unit cap for the manufacturer, and the total cap allocated should be $\phi_{M}D\left(t\right)$. If the total carbon emission at time t exceeds this cap, the manufacturer can sell the rest of the quota in the market to acquire extra revenue. Hence, the objective functionals of the manufacturer and retailer in this scenario are expressed as: (11)$$\pi_{M}^{N}=\int_{0}^{\infty }e^{-rt}\left[\rho_{M}D\left(t\right)+p_{0}\left[\phi_{B}-\left(\phi -\lambda I_{M}\right)\right]D\left(t\right)-\frac{1}{2}k_{M}I_{M}^{2}\right]dt$$
(12)$$\pi_{R}^{N}=\int_{0}^{\infty }e^{-rt}\left[\rho_{R}D\left(t\right)-\frac{1}{2}k_{R}A_{R}^{2}\right]dt$$

After solving the optimization problem in this scenario, we produced the following propositions about optimal decisions and value functions of both supply chain members in this scenario. The proof of propositions are shown in the Appendix A.

**Proposition** **1.***When the favorable event will not happen, the optimal decisions of the manufacturer and the retailer are:*(13)$$I_{M}^{N}=\frac{p_{0}\lambda \left(a+\theta G^{N}\right)}{k_{M}}$$(14)$$A_{R}^{N}=\frac{\gamma \rho_{R}\theta }{k_{R}\left(r+\delta \right)}$$*the time trajectory of the goodwill is:*(15)$$G^{N}\left(t\right)=\left(G_{0}-\frac{\gamma^{2}d_{1}}{\delta k_{R}}\right)e^{-\delta t}+\frac{\gamma^{2}d_{1}}{\delta k_{R}}$$*the value functions for both supply chain members are:*(16)$$V_{M}^{N}\left(G\right)=b_{1}G^{2}+b_{2}G+b_{3}$$(17)$$V_{R}^{N}\left(G\right)=d_{1}G+d_{2}$$*and the parameters in the optimal decisions and value functions are:*(18)$$b_{1}=\frac{{\left(p_{0}\lambda \theta \right)}^{2}}{2k_{M}\left(r+2\delta \right)}$$(19)$$b_{2}=\frac{\theta \Delta }{\left(r+\delta \right)}+\frac{\theta a{\left(p_{0}\lambda \right)}^{2}}{k_{M}\left(r+\delta \right)}+\frac{2\gamma^{2}b_{1}d_{1}}{k_{R}\left(r+\delta \right)}$$(20)$$b_{3}=\frac{a\Delta }{r}+\frac{\gamma^{2}b_{2}d_{1}}{rk_{R}}+\frac{{\left(ap_{0}\lambda \right)}^{2}}{2rk_{M}}$$(21)$$d_{2}=\frac{\rho_{R}a}{r}+\frac{{\left(\gamma d_{1}\right)}^{2}}{2rk_{R}}$$*where*$\Delta =\rho_{M}-p_{0}\phi +p_{0}\phi_{B}$.

We can learn from Proposition 1 that the optimal carbon reduction decision comes from its contribution to the reduction of the unit carbon emission, which therefore increases the revenue acquired by the manufacturer during the carbon trading process. We also find that the manufacturer’s decision is feedback with regard to the state variable G(t). Hence, the change of its value is influenced by the dynamics of goodwill, and it can also be found that $\partial I_{M}^{N}/\partial G>0$, which indicates that the manufacturer should increase his carbon reduction effort when the goodwill increases since a higher level of goodwill means higher market demand, which therefore incurs higher carbon emissions at time t. Meanwhile, the retailer’s optimal decision comes from its contribution to the accumulation of goodwill, which is closely related to the market demand. 

We further analyze the impacts of key parameters upon the optimal decisions in the scenario without consideration of a favorable event, and the results are shown in the following corollaries.

**Corollary** **1.***The manufacturer’s carbon reduction effort*$I_{M}^{N}$*is positively correlated with parameters*γ*,*$\rho_{R}$*,*θ*, and*λ*, and negatively correlated with*δ*and*$k_{M}$*. The retailer’s marketing decision*$A_{R}^{N}$*is also positively correlated with parameters*γ*,*$\rho_{R}$*, and*θ*, and negatively correlated with*δ*and*$k_{R}$.

Since the manufacturer’s carbon reduction decision is a feedback strategy with regard to the goodwill level, the manufacturer should enhance the carbon reduction effort with the increase of goodwill level. Meanwhile, the goodwill level is totally dependent upon the marketing effort, and the marketing effort increases with parameters γ, $\rho_{R}$, and θ, and decreases with parameter δ. Hence, the carbon reduction effort of the manufacturer should also increase with parameters γ, $\rho_{R}$, and θ, and decrease with parameter δ. Meanwhile, the marginal effectiveness and cost parameters are also important factors influencing the optimal decisions of the manufacturer and the retailer.

**Corollary** **2.***The manufacturer’s optimal carbon reduction decision*$I_{M}^{N}$*is positively correlated with the carbon emission quota price*$p_{0}$*. The retailer’s optimal marketing decision*$A_{R}^{N}$*is independent of the carbon emission quota price*$p_{0}$.

Corollary 2 indicates that the manufacturer should enhance the carbon reduction effort when the carbon trading price $p_{0}$ increases. This is because, as the price increases, decreasing the carbon emissions can help the manufacturer gain more revenue during the carbon trading process. At the same time, the retailer’s marketing decision is not affected by the value of the price.

**Corollary** **3.**
*The influence of goodwill on carbon emissions can be expressed as:*

*when*

ϕ≥Ω2

*, then*

$\partial E^{N}\left(t\right)/\partial G^{N}\left(t\right)\geq 0$

*; when*

$\Omega_{1}\leq \phi <\Omega_{2}$

*, then*

$\partial E^{N}\left(t\right)/\partial G^{N}\left(t\right)\leq 0$

*where*$\Omega_{1}=p_{0}\lambda^{2}\left(a+\theta G^{N}\right)/k_{M}$*,*$\Omega_{2}=2p_{0}\lambda^{2}\left(a+\theta G^{N}\right)/k_{M}$.


The influence of goodwill on the carbon emissions of the system can be divided into two aspects. For one thing, the increase of goodwill will contribute to the increase of market demand, which will enhance the carbon emission level according to Equation (3). Meanwhile, the optimal carbon reduction effort is the feedback strategy, with regard to the goodwill level, indicating that a higher goodwill level will enhance the carbon reduction effort. The result of Corollary 3 shows that if the value of ϕ is relatively large, then the carbon emissions will increase when the goodwill level increases, but if the value of ϕ is relatively low, then the increase of the goodwill will contribute to a decrease of emissions.

## 5. Equilibrium Results with the Envisioning of a Favorable Event

In this section, we incorporate the impact of a favorable event in our analysis (superscript IM is used to denote this scenario). Since the event occurring at time T will elevate the goodwill level, the goodwill will not be continuous. Hence, the event splits the whole planning period into two regimes, which are pre-event and post-event regimes (we use subscript i=1,2 to represent the different regimes). Supply chain members should make optimal decisions at each instant t in both pre- and post-event regimes to maximize their long-term expected profits. By applying the backward induction principle [24,25,62,63,64], we start with the post-event regime to solve this optimization problem.

### 5.1. Equilibrium Results in the Post-Event Regime 

We use $G_{2}^{IM}\left(t\right)$ to represent the goodwill evolving in the second regime, which starts with the goodwill level $G_{2}^{IM}\left(T^{+}\right)$ higher than $G_{1}^{IM}\left(T^{-}\right)$ due to the occurrence of the favorable event. The following proposition characterizes the equilibrium results in the post-event regime.

**Proposition** **2.***Equilibrium decisions of both members in the post-event regime are given by:*(22)$$I_{M2}^{IM}=\frac{p_{0}\lambda \left(a+\theta G_{2}^{IM}\right)}{k_{M}}\;$$(23)$$A_{R2}^{IM}=\frac{\gamma_{2}\rho_{R}\theta }{k_{R}\left(r+\delta \right)}\;$$*the time trajectory of the goodwill after the event is given by:*(24)$$G_{2}^{IM}\left(t\right)=\frac{\gamma_{2}^{2}i_{1}}{\delta k_{R}}+\left[\left(1+\varepsilon \right)G_{1}^{IM}\left(T^{-}\right)-\frac{\gamma_{2}^{2}i_{1}}{\delta k_{R}}\right]e^{-\delta \left(t-T\right)},t\geq T\;$$*the value functions for both supply chain members are given by:*(25)$$V_{M2}^{IM}\left(G\right)=h_{1}G_{2}^{2}+h_{2}G_{2}+h_{3}\;$$(26)$$V_{R2}^{IM}\left(G\right)=i_{1}G_{2}+i_{2}\;$$*and the parameters in the equilibrium decisions and value functions are given by:*(27)$$h_{1}=\frac{{\left(p_{0}\lambda \theta \right)}^{2}}{2k_{M}\left(r+2\delta \right)}\;$$(28)$$h_{2}=\frac{\theta \left[k_{M}\Delta +a{\left(p_{0}\lambda \right)}^{2}\right]}{k_{M}\left(r+\delta \right)}+\frac{2\gamma_{2}^{2}h_{1}i_{1}}{k_{R}\left(r+\delta \right)}\;$$(29)$$h_{3}=\frac{k_{R}a\Delta +\gamma_{2}^{2}h_{2}i_{1}}{rk_{R}}+\frac{{\left(ap_{0}\lambda \right)}^{2}}{2rk_{M}}\;$$(30)$$i_{1}=\frac{\rho_{R}\theta }{r+\delta }\;$$(31)$$i_{2}=\frac{a\rho_{R}}{r}+\frac{{\left(\gamma_{2}i_{1}\right)}^{2}}{2rk_{R}}\;$$*where*$\Delta =\rho_{M}-p_{0}\phi +p_{0}\phi_{B}$.

After comparing the equilibrium results in this scenario with those with no favorable event, it can be found that the equilibrium results in the two scenarios having a similar structure, indicating that the parameters, such as the emission quota price $p_{0}$, have similar influence upon the optimal carbon reduction and marketing decisions. However, the manufacturer’s carbon reduction decision in the post-event regime is different from that with no favorable event. This is because the manufacturer’s decision is goodwill-state dependent and the decision the manufacturer makes after the event is based on the goodwill, starting with initial level $G_{2}^{IM}\left(T^{+}\right)$, which equals $\left(1+\varepsilon \right)G_{1}^{IM}\left(T^{-}\right)$ according to the time trajectory of goodwill in the proposition above. Conversely, in the scenario with no favorable event, the goodwill is continuous during the whole planning period. The goodwill in different scenarios evolves in different trajectories, resulting in the difference in carbon reduction decisions in different scenarios.

### 5.2. Equilibrium Results in the Pre-Event Regime

Based on the equilibrium results in the post-event regime, we can further derive the equilibrium results in the pre-event regime. The following proposition characterizes the equilibrium results in the post-event regime.

**Proposition** **3.***Equilibrium decisions of both members in the pre-event regime are given by:*(32)$$I_{M1}^{IM}=\frac{p_{0}\lambda \left(a+\theta G_{1}^{IM}\right)}{k_{M}}$$(33)$$A_{R1}^{IM}=\frac{\gamma_{1}\rho_{R}\theta \left[r+\delta +\chi \left(1+\varepsilon \right)\right]}{k_{R}\left(r+\delta \right)\left(r+\chi +\delta \right)}$$*the time trajectory of goodwill before the event is given by:*(34)$$G_{1}^{IM}\left(t\right)=\left(G_{0}-\frac{\gamma_{1}^{2}g_{1}}{\delta k_{R}}\right)e^{-\delta t}+\frac{\gamma_{1}^{2}g_{1}}{\delta k_{R}},G_{1}^{IM}\left(0\right)=G_{0},0\leq t<T\;$$*the value functions for both supply chain members are given by:*(35)$$V_{M1}^{IM}\left(G\right)=f_{1}G_{1}^{2}+f_{2}G_{1}+f_{3}\;$$(36)$$V_{R1}^{IM}\left(G\right)=g_{1}G_{1}+g_{2}\;$$*and the parameters in the equilibrium decisions and value functions are given by:*(37)$$f_{1}=\frac{{\left(p_{0}\lambda \theta_{1}\right)}^{2}}{2k_{M}\left(r+\chi +2\delta \right)}+\frac{h_{1}\chi {\left(1+\varepsilon \right)}^{2}}{r+\chi +2\delta }\;$$(38)$$f_{2}=\frac{\theta_{1}\Delta +h_{2}\chi \left(1+\varepsilon \right)}{r+\chi +\delta }+\frac{2\gamma_{1}^{2}f_{1}g_{1}}{k_{R}\left(r+\chi +\delta \right)}+\frac{\theta_{1}a{\left(p_{0}\lambda \right)}^{2}}{k_{M}\left(r+\chi +\delta \right)}\;$$(39)$$f_{3}=\frac{a\Delta +\chi h_{3}}{r+\chi }+\frac{{\left(ap_{0}\lambda \right)}^{2}}{2k_{M}\left(r+\chi \right)}+\frac{\gamma^{2}f_{2}g_{1}}{k_{R}\left(r+\chi \right)}\;$$(40)$$g_{1}=\frac{\rho_{R}\theta_{1}+\chi i_{1}\left(1+\varepsilon \right)}{r+\chi +\delta }\;$$(41)$$g_{2}=\frac{\rho_{R}a+\chi i_{2}}{r+\chi }+\frac{{\left(\gamma_{1}g_{1}\right)}^{2}}{2k_{R}\left(r+\chi \right)}\;$$*where*$\Delta =\rho_{M}-p_{0}\phi +p_{0}\phi_{B}$.

It worth noting that the decisions made by supply chain members in the pre-event regime bear a similar structure compared with those in the post-event regime. Hence, decisions before and after the event have similar properties. For example, marketing decision $A_{R1}^{IM}$ increases in λ and $k_{M}$. Carbon reduction decision $I_{M1}^{IM}$ increases in λ and $p_{0}$, and decreases in $k_{M}$. We can also learn from Proposition 3 that, different from the optimal decision after the event, the retailer’s marketing decision before the event is influenced by the value of occurrence rate χ and expansion rate ε, indicating that, when the favorable event is likely to happen, the retailer should evaluate the possibility and impact of the event when making the decision at the initial time to maximize his long-term profit. In addition, if χ=0, which means the favorable event will not happen, the equilibrium marketing decision will be identical to that in Proposition 1. The manufacturer should also take the favorable event and its influence into consideration at the beginning, for his decision is dependent upon goodwill which will be affected by χ and ε through marketing effort. After further analyzing the impacts of χ and ε on equilibrium decisions, we can have the following corollaries.

**Corollary** **4.***The retailer’s marketing effort*$A_{R1}^{IM}$*and the manufacturer’s carbon reduction effort*$I_{M1}^{IM}$*before the favorable event increases with the increase of the occurrence rate*χ.

When the occurrence rate increases, the probability of the favorable event increases, and the event happens earlier according to the analysis in 3.2. The retailer should enhance his marketing effort to accelerate the accumulation of goodwill before the event with the increase of probability, even if it may increase the carbon emission due to the higher level of goodwill and market demand. This is because the event will elevate the goodwill by expansion rate ε, and makes the marketing investment of the retailer before the event more effective in the accumulation of goodwill. Hence, when the favorable event is likely to happen and the occurrence rate increases, the retailer may disregard the carbon emission problem in exchange for higher marketing influence. Meanwhile, since the manufacturer’s decision is state-dependent, he should also increase his carbon reduction effort in the pre-event regime with higher marketing effort and goodwill level. Under the cap-and-trade regulation, a high level of goodwill has both negative and positive influence on the manufacturer’s profit, causing the manufacturer to balance the revenue both from the product and the carbon trading market.

**Corollary** **5.***The retailer’s marketing effort*$A_{R1}^{IM}$*and the manufacturer’s carbon reduction effort*$I_{M1}^{IM}$*before the favorable the favorable event increases with the increase of the expansion rate*ε.

The expansion rate measures the influence of the favorable event on goodwill. When the parameter increases, the retailer should increase his marketing effort to establish a higher goodwill level before the event. At the same time, the manufacturer should increase his carbon reduction effort as well. Hence, we can conclude from Corollary 4 and Corollary 5 that the potential favorable event will increase the marketing effort of the retailer and the carbon reduction effort of the retailer.

**Corollary** **6.**
*The influence of the occurrence rate*

χ

*and the expansion rate*

ε

*on carbon emissions before the favorable event can be expressed as:*

*when*

ϕ≥Ω2

*, then*

$\partial E_{1}^{IM}\left(t\right)/\partial \chi \geq 0$

*,*

$\partial E_{1}^{IM}\left(t\right)/\partial \varepsilon \geq 0$

*;*

*when*

$\Omega_{1}\leq \phi <\Omega_{2}$

*, then*

$\partial E_{1}^{IM}\left(t\right)/\partial \chi <0$

*,*

$\partial E_{1}^{IM}\left(t\right)/\partial \varepsilon <0$

*where*Ω1=p0λ2(a+θGN)/kM*,*Ω2=2p0λ2(a+θGN)/kM.


According to Corollary 3, it can be found in this scenario that when $\phi \geq \Omega_{2}$, we will have $\partial E_{1}^{IM}\left(t\right)/\partial G^{IM}\geq 0$ and when $\Omega_{1}\leq \phi <\Omega_{2}$, we will have $\partial E_{1}^{IM}\left(t\right)/\partial G^{IM}\leq 0$. Meanwhile, the retailer’s marketing effort positively contributes to the goodwill and the marketing effort before the favorable event will increase with the increase of occurrence rate χ and the expansion rate ε. Hence, when the parameters increase, the goodwill before the event will also increase, indicating that we will have $\partial G^{IM}/\partial \chi \leq 0$ and $\partial G^{IM}/\partial \varepsilon \leq 0$. According to the chain rule, we can acquire the results of Corollary 6. Hence, the influence of parameters on carbon emissions will also depend on the value of the unit carbon emission. When the unit carbon emission is relatively high, the carbon emission before the favorable event will increase with the increase of occurrence rate χ and the expansion rate ε; however, when the unit carbon emission is relatively low, the carbon emission before the favorable event will decrease with the increase of occurrence rate χ and the expansion rate ε. Hence, the favorable event can help lower the carbon emissions under certain circumstances.

## 6. Comparison Analysis 

We compare the optimal results of different scenarios to conduct further analysis on the impacts of a favorable event in this section. We first compare the decisions with the impacts of the favorable event with those with no event, and the results of the comparison are shown in the following propositions.

**Proposition** **4.***The optimal carbon reduction decision with and without the favorable event can be compared as*$I_{M}^{N}<I_{M1}^{IM}$*,*$I_{M}^{N}<I_{M2}^{IM}$*; the marketing decision with and without the event can be compared as*$A_{R}^{N}<A_{R1}^{IM}$*,*$A_{R}^{N}<A_{R2}^{IM}$*; the optimal profits of the manufacturer and the retailer with and without event can be compared as*$V_{M}^{N}\left(G\right)<V_{M1}^{IM}\left(G\right)$*,*$V_{R}^{N}\left(G\right)<V_{R1}^{IM}\left(G\right)$.

The result of Proposition 4 indicates that the presence of the potential favorable event will encourage the retailer to enhance his marketing effort before the event, which therefore can enhance the goodwill level in the pre-event regime. With the increase of goodwill level in the pre-event regime, the manufacturer will also increase the carbon reduction effort before the event. Meanwhile, the marketing effort level after the event is also higher than that with no favorable event, due to the increase of the marginal effectiveness after the event, which results in the enhancement of goodwill. Influenced by the enhanced goodwill level, the manufacturer’s carbon reduction effort level after the event is also higher than that with no favorable event. Jointly affected by the marketing and carbon reduction efforts, the profits of the manufacturer and the retailer with the influence of the favorable event are bigger than those with no event. Hence, we can draw the conclusion from Proposition 4 that the presence of a favorable event will enhance the marketing and carbon reduction efforts in both pre- and post-event regimes, as well as the profits of the manufacturer and the retailer.

**Proposition** **5.**
*The optimal marketing effort before and after the event can be compared as:*

*when*

$\gamma_{2}/\gamma_{1}<\Omega_{3}$

*, then*

$A_{R1}^{IM}>A_{R2}^{IM}$

*; when*

$\gamma_{2}/\gamma_{1}\geq \Omega_{3}$

*, then*

$A_{R1}^{IM}\leq A_{R2}^{IM}$

*where*$\Omega_{3}=\frac{r+\delta +\chi \left(1+\varepsilon \right)}{r+\chi +\delta }$.


According to the assumptions in the previous section, the favorable event will not only enhance the goodwill level, but will also enhance the marketing effectiveness on the goodwill. Hence, whether the retailer should increase his marketing effort after the favorable event depends on the value of marginal effectiveness of marketing before and after the event. Proposition 5 shows that when $\gamma_{2}/\gamma_{1}$ is smaller than the threshold $\Omega_{3}$, the retailer should reduce his marketing effort in the post-event regime; however, when $\gamma_{2}/\gamma_{1}$ exceeds the threshold $\Omega_{3}$, the retailer should enhance the marketing effort in this regime. Hence, if the favorable event exerts great influence on the marketing effectiveness, the retailer should increase his investment in marketing after the event. Meanwhile, the first order partial derivative of $\Omega_{3}$ with regard to the occurrence rate χ and the expansion rate ε are expressed as follows:(42)$$\frac{\partial \Omega_{3}}{\partial \chi }=\frac{\varepsilon \left(r+\delta \right)}{{\left(r+\chi +\delta \right)}^{2}}>0,\frac{\partial \Omega_{3}}{\partial \varepsilon }=\frac{\chi }{r+\chi +\delta }>0$$

The results above show that when the occurrence rate χ and the expansion rate ε increase, the threshold $\Omega_{3}$ will also increase and $\gamma_{2}/\gamma_{1}$ is less likely to be bigger than $\Omega_{3}$, indicating that the retailer is more likely to invest more in the pre-event regime when these parameters increase. However, when the occurrence rate and the expansion rate decrease, the threshold $\Omega_{3}$ will decrease and the retailer is less likely to invest more in the pre-event regime. In fact, when the occurrence rate increases, the probability that the favorable event occurs increases and the pre-event regime becomes shorter, which urges the retailer to enhance the marketing effort before the event and accumulate the goodwill level. This is because the favorable event elevates the goodwill level based on the level before the event, indicating that a higher level of goodwill before the event is very desirable. Hence, when the occurrence rate increases, the retailer should enhance the marketing investment to try his best to elevate the goodwill level before the favorable event. However, when the occurrence rate decreases, the time that the event occurs will become later, and it becomes less urgent for the retailer to build the goodwill level before the event. Meanwhile, it can be found that when χ→∞, then $\Omega_{3}\to 1+\varepsilon$, and when χ→0, then $\Omega_{3}\to 1$. Hence, the range of the threshold $\Omega_{3}$ is: (43)$$1<\Omega_{3}<1+\varepsilon$$

The value of $\gamma_{2}/\gamma_{1}$ reflects the regime preference of the retailer, and threshold $\Omega_{3}$ is an important index to measure the preference. When $\gamma_{2}/\gamma_{1}$ is less than $\Omega_{3}$ (the marketing effectiveness is not greatly elevated by the event), the marketing investment in the pre-event regime is more efficient, and the retailer should lower the investment after the event. However, when $\gamma_{2}/\gamma_{1}$ is bigger than $\Omega_{3}$ (the effectiveness of marketing effort is substantially elevated by the event), the marketing investment in the post-event regime is more efficient and the retailer should enhance the marketing investment after the event. However, the value of $\Omega_{3}$ has an upper and lower limit. Hence, according to the value of $\gamma_{2}/\gamma_{1}$, we can further discuss the regime preference in the following three scenarios.

**Scenario** **1:**When $\gamma_{1}=\gamma_{2}$, then $A_{R1}^{IM}>A_{R2}^{IM}$ for ∀χ>0.

$\gamma_{1}=\gamma_{2}$ means that the favorable event exerts no influence on the marketing effectiveness on goodwill, even though the event will elevate the goodwill level, indicating that the goodwill in the post-event regime evolves in the same rule as the goodwill in the pre-event regime. Hence, in this scenario, the marketing investment before the favorable event is more efficient than that after the event for the reason that the goodwill level in the pre-event regime can be elevated by the favorable event (the favorable event elevates the goodwill level by ε, which is equivalent to increase the efficiency of investment before the event by ε with no increase of actual investment), but the goodwill level after the event will not be elevated by another favorable event according to the assumption. Hence, the retailer invests more in the pre-event regime. Meanwhile, in this scenario, the marketing effort level in the post-event regime is identical to that with no favorable event, namely $A_{R2}^{IM}=A_{R}^{N}$.

**Scenario** **2:**When $\gamma_{2}/\gamma_{1}>1+\varepsilon$, then $A_{R1}^{IM}\leq A_{R2}^{IM}$ for ∀χ>0.

This scenario features the fact that the favorable event can not only elevate the goodwill level but will also substantially enhance the marketing effectiveness on goodwill. In this scenario, the retailer should increase the marketing investment after the event for the investment in this regime is more efficient due to the increase of marketing effectiveness. Hence, if the long-term effect (effect on the marginal effectiveness) outweighs the short-term effect (elevation of the goodwill due to the favorable event), the retailer should put more emphasis on the post-event regime and allocate more investment in this regime.

**Scenario** **3:**When $1<\gamma_{2}/\gamma_{1}\leq 1+\varepsilon$, then $A_{R1}^{IM}>A_{R2}^{IM}$ for $\chi >\tilde{\chi }$; and $A_{R1}^{IM}\leq A_{R2}^{IM}$ for $\chi \leq \tilde{\chi }$ where $\tilde{\chi }=\frac{\left(r+\delta \right)\left(\gamma_{2}-\gamma_{1}\right)}{\gamma_{1}\left(1+\varepsilon \right)-\gamma_{2}}$.

Different from the Scenario 1 and Scenario 2, in this scenario, even though the favorable event can enhance the marginal effectiveness, the marginal effectiveness is not substantially enhanced, similar to the situation in Scenario 2. $\gamma_{2}/\gamma_{1}\leq 1+\varepsilon$ means that $\left(\gamma_{2}-\gamma_{1}\right)/\gamma_{1}\leq \varepsilon$, indicating that the increment rate of $\gamma_{1}$ due to the favorable event is less than ε. Under this circumstance, the retailer should examine the value of the occurrence rate. If the occurrence rate is relatively high ($\chi >\tilde{\chi }$), it is unnecessary for the retailer to maintain a high marketing effort level after the favorable event. However, if the occurrence rate is relatively low, then the retailer should enhance the marketing effort level after the event. 

Hence, we can conclude from these three scenarios that the retailer should invest more in the regime with high investment efficiency, and the determining factor are the occurrence rate and marginal effectiveness before and after the event. When the occurrence rate is high, the retailer should be more preferable to the pre-event regime, and when the marginal effectiveness in the post-event regime is high, then the retailer should put more emphasis on the post-event regime. 

**Proposition** **6.**
*The optimal carbon reduction decision before and after the event can be compared as:*

(44)
$$I_{M1}^{IM}\left(G\left(T^{-}\right)\right)<I_{M2}^{IM}\left(G\left(T^{+}\right)\right)$$



Since the carbon reduction decision is feedback with regard to the goodwill level, which will be directly elevated by the favorable event, the carbon reduction effort right after the event must be higher than the effort level right before the event. However, the carbon reduction level in the post-event regime may not always be bigger than that in the pre-event regime. This is because when $\gamma_{2}/\gamma_{1}<\Omega_{3}$, we have $A_{R1}^{IM}>A_{R2}^{IM}$, indicating that the goodwill level may be lower than that before the event, and we will have $I_{M1}^{IM}>I_{M2}^{IM}$ after a certain instant. However, when $\gamma_{2}/\gamma_{1}\geq \Omega_{3}$, we have $A_{R1}^{IM}\leq A_{R2}^{IM}$, indicating that the carbon reduction effort after the event will always be bigger than that before the event. 

## 7. Numerical Analysis 

In this section, we numerically analyze the favorable event and confirm the conclusions aforementioned. The values of the basic parameters are given as follows: $G_{0}=10$; r=0.1; $\gamma_{1}=1$; $\gamma_{2}=10$; δ=0.5; a=10; β=1; θ=2; ϕ=3; λ=0.1; $k_{M}=1$; $k_{R}=1$; $\rho_{M}=6$; $\rho_{R}=2$; $p_{0}=2$. We first analyze the impacts of the events, based on which we will further discuss the decisions before and after the event.

### 7.1. Impacts of the Favorable Event 

#### 7.1.1. Impacts of the Favorable Event on the Marketing Decision in the Pre-Event Regime

In this subsection, we first study the impacts of the favorable event on the marketing effort in the pre-event regime, as are shown in the following Figure 1, Figure 2 and Figure 3. According to the result in Proposition 3, if the favorable event is likely to happen, the retailer should take the occurrence rate and expansion rate into consideration when making decisions before the event, indicating that the retailer should make adjustment of his decision according to the parameters. The following figures show that when the occurrence rate or the expansion rate increase, the retailer should enhance the marketing effort in the pre-event regime to further enhance the goodwill level before the event. This is because the favorable event can elevate the goodwill level by the expansion rate, which is equivalent to enhancing the investment efficiency of the retailer before the event. After comparison with the results in research concerning the study of brand crisis [24,25], we found that the results in this study are quite opposite to those in extant studies. This is because the brand crisis will damage the goodwill and therefore reduce the investment efficiency before the crisis. Hence, the strategy the company should adopt depends on what kind of event the company may encounter. 

#### 7.1.2. Impacts of the Favorable Event on the Goodwill 

Based on the analysis of marketing effort with the influence of the favorable event, we further discuss the influence of this event on the goodwill level. According to Equation (1), the goodwill is dynamically affected by the marketing effort, indicating that the influence of the favorable event on the goodwill entirely depends on how the event influences the marketing effort. Since the retailer will increase the marketing effort before the event with the increase of the occurrence rate and expansion rate, the goodwill level before the event will also increase when the parameters increase. For example, in Figure 4, with a certain value of expansion rate, the goodwill level before the event is higher with a higher value of occurrence rate, indicating that the goodwill with a higher value of the occurrence rate will also have a higher level right after the event. However, due to the fact that the favorable event will not affect the marketing effort after the event, the goodwill level with a different occurrence rate will eventually evolve to the same steady state. Figure 5 shows the influence of the expansion rate on the goodwill level. It worth noting from Figure 5 that the goodwill after the event decreases with time. This is because the goodwill level before the event may be elevated to a level higher than the steady state, indicating that the goodwill will decrease to the steady state. 

#### 7.1.3. Impacts of the Favorable Event on the Carbon Reduction Effort

Figure 6 and Figure 7 exhibit the impacts of the favorable event on the manufacturer’s carbon reduction effort. According to the results of Propositions 2 and 3, the optimal carbon reduction effort is feedback strategy regarding the goodwill level. Hence, the evolution of the optimal carbon reduction effort before and after the event is quite dependent on the dynamics of the goodwill. Since the goodwill level will increase when the occurrence rate or the expansion rate increases, the manufacturer should also increase his carbon reduction effort when the parameters increase. The potential favorable event will enhance the demand before and after the event by stimulating marketing effort and elevating goodwill, which will eventually enhance the carbon emission level. To offset the negative effect caused by higher carbon emissions, the manufacturer should also enhance the carbon reduction effort. Meanwhile, compared to extant literatures about the study of the dynamic carbon reduction effort, such as [40,41], the favorable event is likely to happen, and the carbon reduction effort is not continuous during the whole planning period and will be affected by the probability of the event. 

#### 7.1.4. Impacts of the Favorable Event on the Carbon Emission

It can be learned from the analysis in Section 2 that the carbon emission level is jointly determined by the marketing effort, which will elevate the goodwill and market demand, as well as the carbon reduction effort. Meanwhile, according to the result in Section 7.1.1 and Section 7.1.3, the increase of occurrence rate and expansion rate will enhance the market effort, as well as the carbon reduction effort. However, the increase of the marketing effort will help to accumulate carbon emissions, while the increase of the carbon reduction effort will help to reduce the carbon emissions. Hence, under the opposite influences of the supply chain members’ effort, the impacts of the favorable event on the carbon emissions are uncertain. According to the result of Corollary 6, the impacts of the occurrence rate and expansion rate depend on the value of ϕ. If the value of ϕ is relatively low ($\Omega_{1}\leq \phi <\Omega_{2}$), then the emission level will decease with the increase of the occurrence rate and expansion rate, as is shown in Figure 8 and Figure 9. However, if the value of ϕ is relatively large ($\phi \geq \Omega_{2}$), then the emission level will increase with the increase of these parameters, as is shown in Figure 10 and Figure 11. This is because when the value of ϕ is low, the marketing effort exerts a small influence in enhancing the total emission level, and the carbon reduction effort is effective in curbing the carbon emissions. However, if the value of ϕ is high, then the marketing effort exerts a big influence in enhancing the total emissions level, and the carbon emissions level will increase when both the marketing effort and the carbon reduction effort increase. 

### 7.2. Comparsion of Decisions and Goodwill before and after the Event

Proposition 5 compares and analyzes the retailer’s marketing decisions before and after the favorable event, and this section further analyzes the results of Proposition 5 numerically. Figure 12 shows the marketing decision before and after the event in Scenario 1, and Figure 13 is the corresponding goodwill level. In this scenario, even though the favorable can elevate the goodwill level, it exerts no long-term influence on the marketing effectiveness ($\gamma_{1}=\gamma_{2}$). Hence, the marketing decision after the event is identical to that with no favorable event. However, if influenced by the favorable event, the marketing effort before the event is biggest. Since the marketing effort before the event is bigger than that with no favorable event, the goodwill level before the event is also bigger than that with no event. Meanwhile, since the marketing effort is identical to that with no event, the steady state of goodwill after the event is also identical to that with no event, even if the goodwill after the event is elevated to a higher level. 

Figure 14 shows the comparison result marketing decisions in the Scenario 2. This scenario features the fact that the favorable event will enhance the marginal effectiveness of the marketing effort to a level much higher than the level before the event ($\gamma_{2}/\gamma_{1}>1+\varepsilon$). This indicates that the long-term influence of the event exceeds the short-term influence and the investment efficiency after the event is greater than that before the event. Hence, the retailer in this scenario should further enhance his investment after the event, as is shown in Figure 14. Figure 15 exhibits the dynamics of goodwill when $\gamma_{2}/\gamma_{1}>1+\varepsilon$. Different from the scenario shown in Figure 13, the steady state of goodwill with a favorable event is higher than that with no favorable event. This is because the marketing effort after the event is bigger than that with no event.

Figure 16 compares the marketing effort in Scenario 3. In this scenario, even though the favorable event exerts long-term influence and enhances the marketing effectiveness, the growth rate of the marginal effectiveness does not exceed ε ($1<\gamma_{2}/\gamma_{1}\leq 1+\varepsilon$). Hence, whether the retailer should enhance the investment or not after the event depends on the value of occurrence rate χ. When the occurrence rate is relatively large ($\chi >\tilde{\chi }$), the retailer should lower the investment after the event and put more emphasis on the pre-event regime. However, when the occurrence rate is relatively low ($\chi \leq \tilde{\chi }$), then the retailer should enhance the investment after the event. Figure 17a,b present the dynamics of the goodwill when $\chi >\tilde{\chi }$ and $\chi \leq \tilde{\chi }$, respectively. 

Hence, it can be found from the analysis above that the presence of the favorable event splits the whole planning period into two regimes, which are pre-event and post-event, respectively. Moreover, the values of the parameters determine the investment efficiency of each regime. The increase of the occurrence rate and expansion rate will enhance the investment efficiency of the pre-event regime, and the increase of the marginal effectiveness after the event will enhance the investment efficiency of the post-event regime. The supply chain members should adjust their investment strategy in different scenarios and invest more in the regime with higher investment efficiency.

## 8. Conclusions 

This paper pays attention to a situation not often discussed previously. In the practical operation process of a supply chain system, the members may encounter a favorable event which will not only elevate the goodwill, which results in an increase of market demand and carbon emissions to a higher level after the event but will also enhance the marketing effectiveness and exert a long-term effect on the dynamics of the goodwill. However, the enhancement of goodwill indicates the increase of carbon emissions, and therefore may suffer loss under the cap-and-trade regulation. Hence, the purpose of this paper is to explore the optimal marketing and carbon reduction decisions with the potential favorable event. To this end, we first use the random occurrence process to depict the favorable event, based on which we construct a differential model of the supply chain system with this event. After solving and analyzing the model, we acquire the following conclusions:

(1) The occurrence of the favorable event splits the whole planning period into two regimes, the pre-event and post-event regimes, respectively, and the supply chain members should make decisions in each regime to maximize the overall profits. Meanwhile, when they make decisions in the pre-event regime, they should take the event into consideration and should adjust their decision with regard to the occurrence rate of the event and expansion rate of the goodwill. When the occurrence rate and the expansion rate increase, the retailer should enhance the marketing effort before the event to try their best to increase the goodwill level before the event. Affected by the increased marketing effort, the goodwill before the event will also increase with the increase of these parameters, and is higher than that with no favorable event. Since the carbon reduction effort is a feedback strategy with regard to the goodwill level, the carbon reduction effort will also increase when the goodwill level increases. Hence, the potential favorable event will elevate the marketing and carbon reduction effort, as well as the goodwill level before the event. 

(2) The occurrence of the favorable event will elevate the goodwill level after the event and therefore will increase the investment efficiency before the event. This is the reason why the retailer should increase the marketing effort before the event. Meanwhile, the event favorable may raise the marketing effectiveness after the event, indicating that investment efficiency after the event may also be higher than that with no event. Hence, the retailer and the manufacturer should fully consider the impacts of the event when deciding whether to increase the investment after the event. If the favorable event will not raise the marketing effectiveness after the event ($\gamma_{1}=\gamma_{2}$), or just slightly raise the effectiveness ($1<\gamma_{2}/\gamma_{1}\leq 1+\varepsilon$), then the investment before the favorable event is still more efficient than that after the event, indicating that the supply chain members should decrease the investment in the post-event regime and allocate more investment in the pre-event regime. However, if the event can considerably raise the marketing effectiveness ($\gamma_{2}/\gamma_{1}>1+\varepsilon$), then the post-regime investment is more efficient and both members should enhance the investment after the event.

(3) The carbon emissions of the system will be increased by the marketing effort and will be reduced by the carbon reduction effort. Hence, the impact of the favorable event on the carbon emissions is uncertain for the event will simultaneously increase both the marketing and the carbon reduction effort. Our analysis reveals that if the value of the unit carbon emissions ϕ is relatively low ($\Omega_{1}\leq \phi <\Omega_{2}$), then the emissions quantity will decease with the increase of the occurrence rate and expansion rate. However, if the value of ϕ is relatively large ($\phi >\Omega_{2}$), then the emissions quantity will increase with the increase of these parameters. This is because when the value of ϕ is low, the marketing effort exerts a small influence in enhancing the emissions quantity and the carbon reduction effort is effective in curbing the carbon emissions. However, if the value of ϕ is high, then the marketing effort exerts a large influence in enhancing the emissions quantity, and the emissions quantity will eventually increase when both the marketing effort and carbon reduction effort increase. Hence, the favorable event may not always help to increase the carbon emissions, even if the event can enhance the goodwill and the marketing demand. From perspective of the government, the favorable event may become a way of reducing the carbon emissions, since, when the value of ϕ is relatively low, the increase of the occurrence rate and expansion rate will decrease the carbon emissions. In this scenario ($\Omega_{1}\leq \phi <\Omega_{2}$), the government can take measures to amplify the influence of the event. For example, they can help the company to advertise the event and enhance the expansion rate of goodwill, which will further decrease the carbon emissions of the supply chain. However, if the unit carbon emissions are relatively large, it is undesirable for the government to advertise the event from a carbon reduction perspective.

## 9. Limitations and Suggestions for Future Research

We also offer some limitations of this paper and propose some suggestions for future research. Firstly, we only study the benchmarking rule in the cap-and-trade regulation. In the future, we can study and compare different mechanisms of the cap-and-trade regulation with consideration of the favorable event. Secondly, we fail to incorporate the pricing problem in this research. In the future, we can further study the impacts of a favorable event and the cap-and-trade regulation on the pricing of supply chain members to see how the members adjust their pricing strategies to optimize their profits. Thirdly, we assume the carbon trading price and carbon quota to be exogenous variables instead of control variables. In the future, if we involve the policymakers as a member of the game, we may discover how the policymakers make decisions on carbon trade price and carbon quota from the perspective of achieving optimal social welfare.

## Figures and Tables

**Figure 1 ijerph-20-04644-f001:**
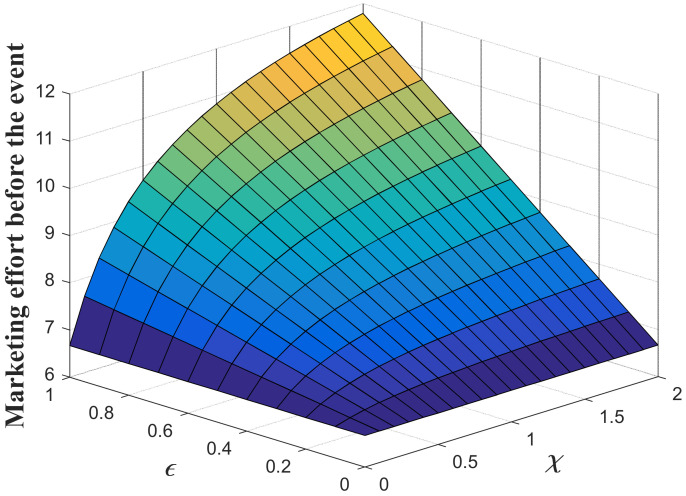
Marketing effort before the favorable event.

**Figure 2 ijerph-20-04644-f002:**
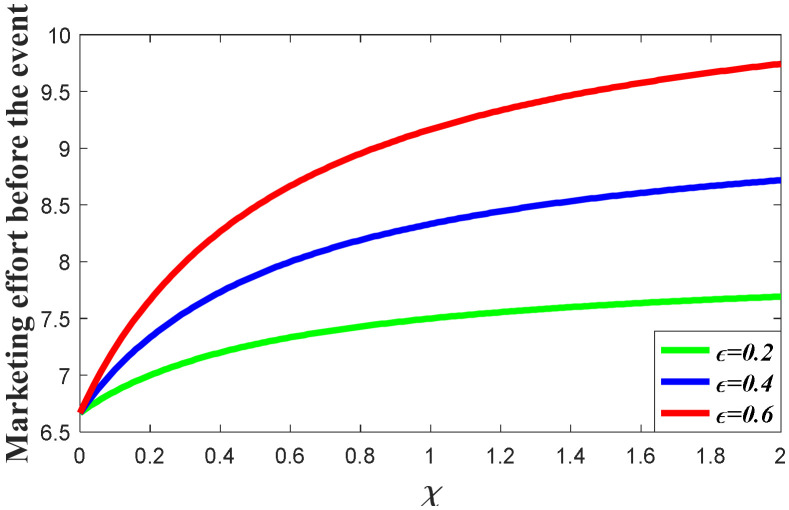
Impacts of ε on marketing decision.

**Figure 3 ijerph-20-04644-f003:**
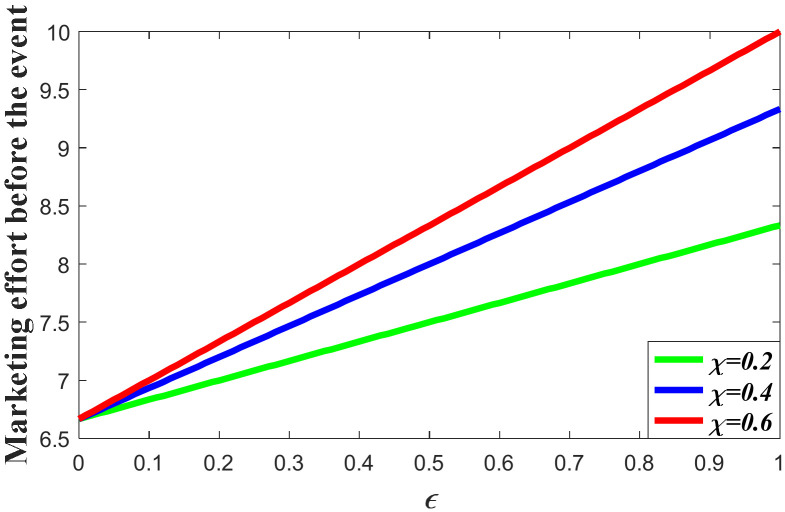
Impacts of χ on marketing decision.

**Figure 4 ijerph-20-04644-f004:**
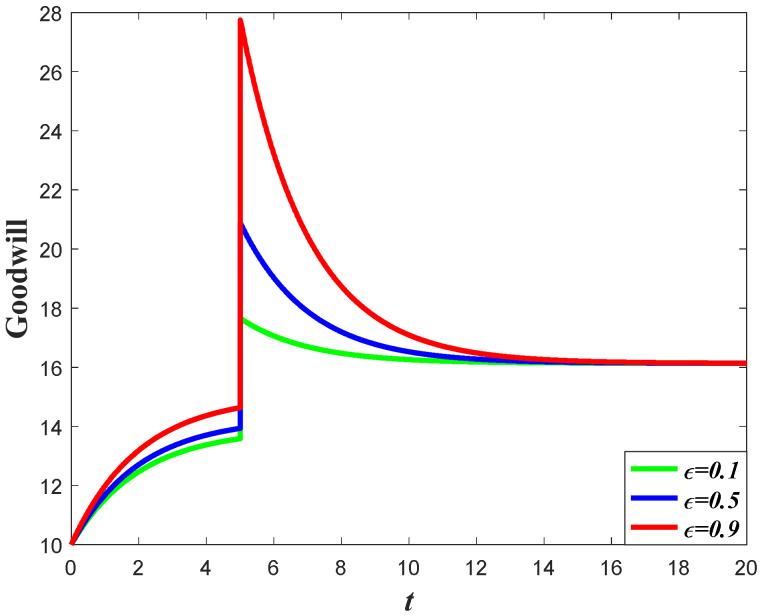
Impacts of ε on goodwill level.

**Figure 5 ijerph-20-04644-f005:**
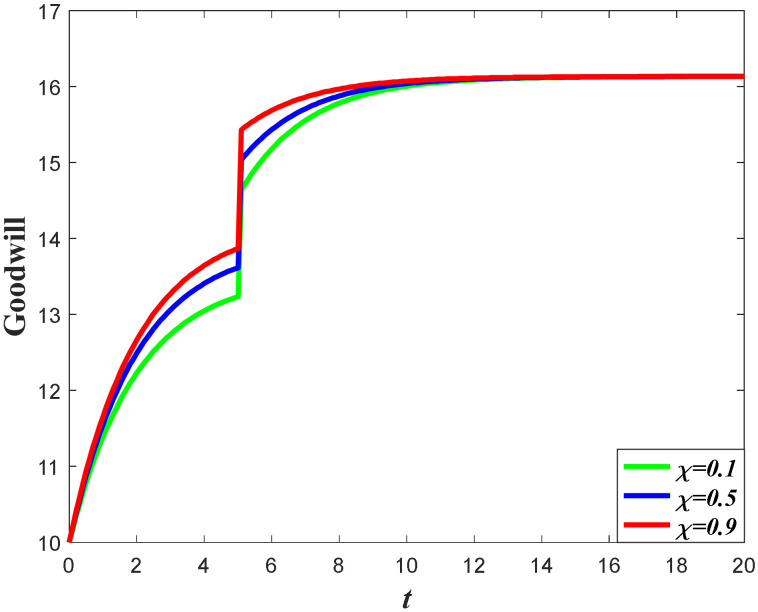
Impacts of χ on goodwill level.

**Figure 6 ijerph-20-04644-f006:**
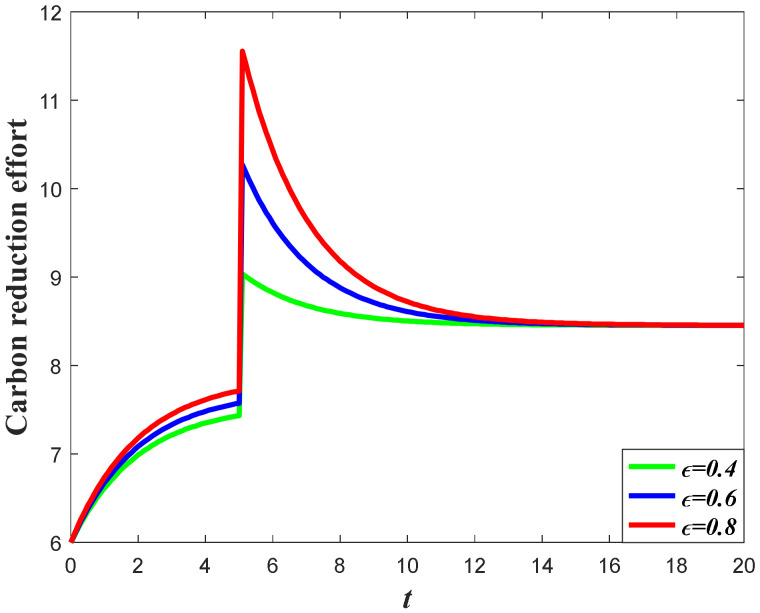
Impacts of ε on carbon reduction effort.

**Figure 7 ijerph-20-04644-f007:**
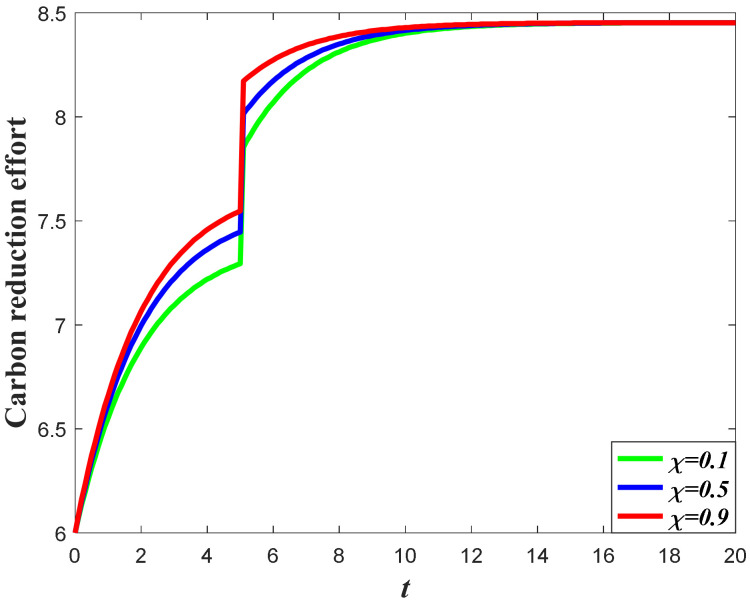
Impacts of χ on carbon reduction effort.

**Figure 8 ijerph-20-04644-f008:**
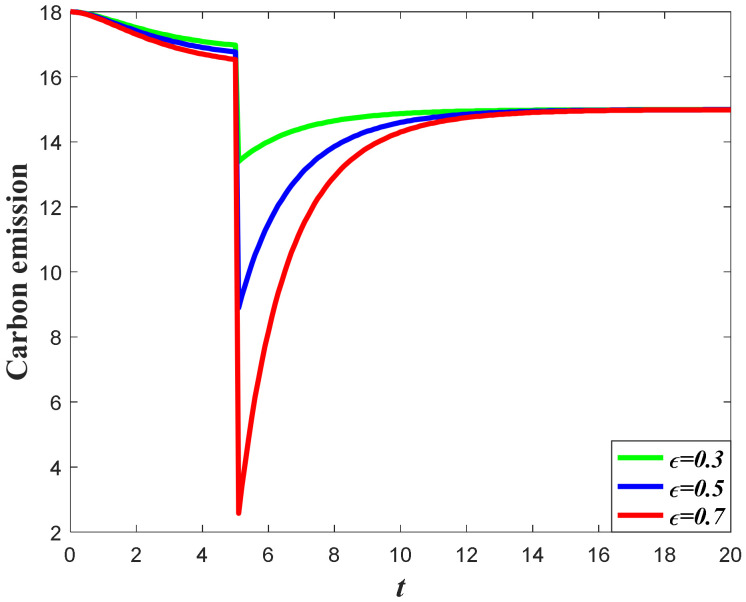
Impacts of ε on carbon emission when $\Omega_{1}\leq \phi <\Omega_{2}$.

**Figure 9 ijerph-20-04644-f009:**
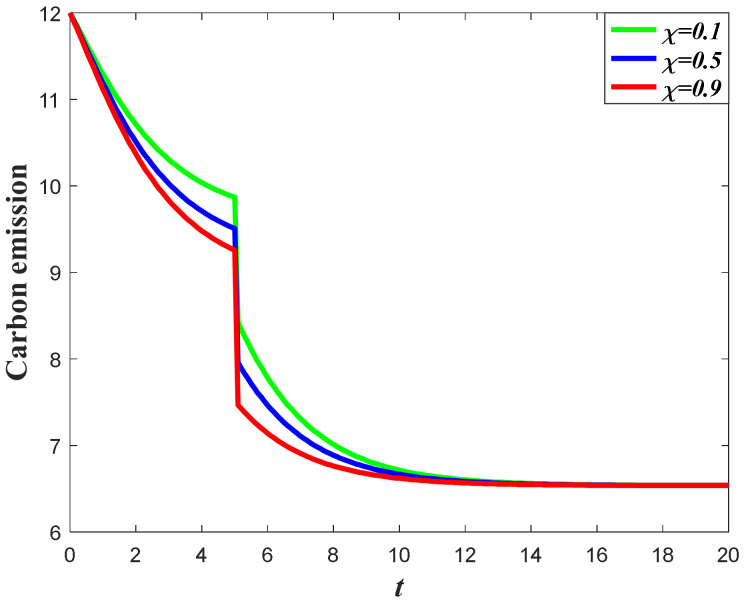
Impacts of χ on carbon emission when $\Omega_{1}\leq \phi <\Omega_{2}$.

**Figure 10 ijerph-20-04644-f010:**
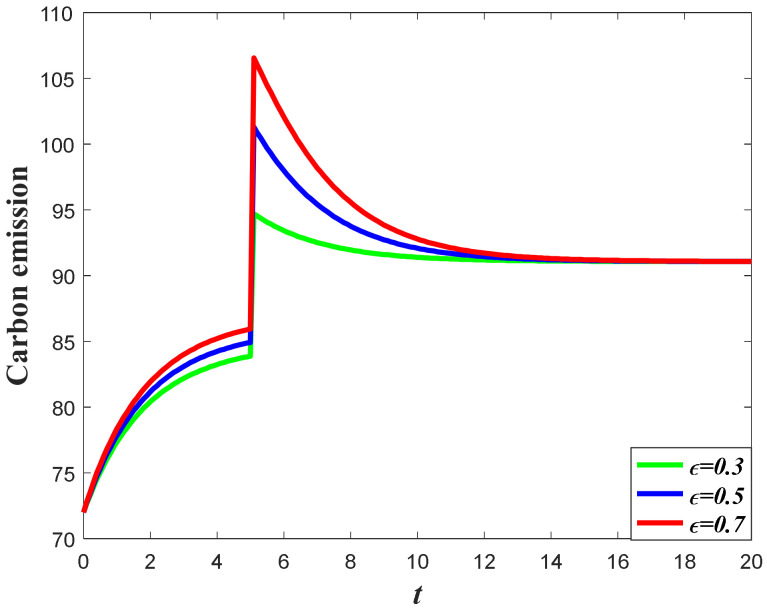
Impacts of ε on carbon emission when $\phi \geq \Omega_{2}$.

**Figure 11 ijerph-20-04644-f011:**
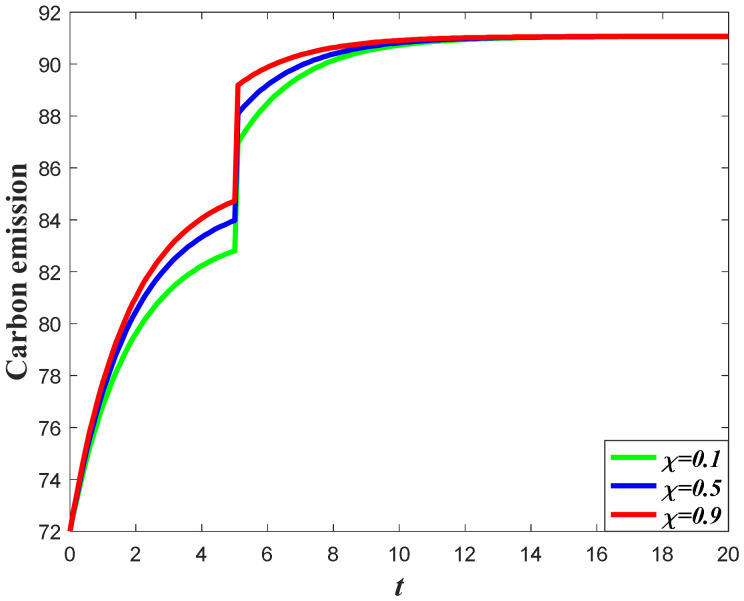
Impacts of χ on carbon emission when $\phi \geq \Omega_{2}$.

**Figure 12 ijerph-20-04644-f012:**
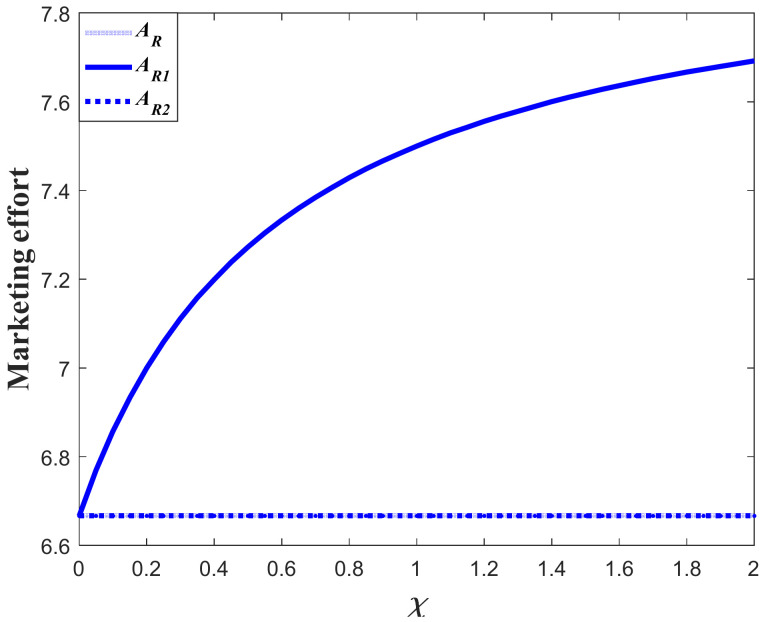
Marketing effort before and after the event (Scenario 1).

**Figure 13 ijerph-20-04644-f013:**
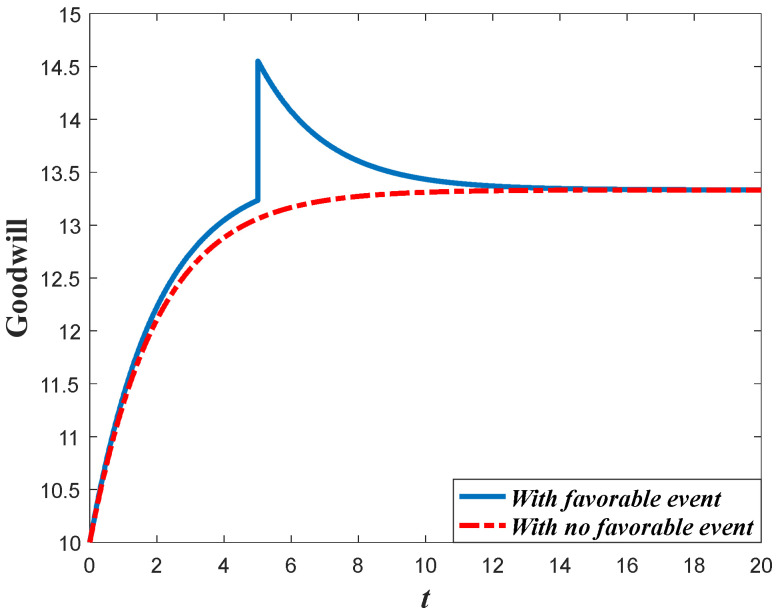
Goodwill level before and after the event (Scenario 1).

**Figure 14 ijerph-20-04644-f014:**
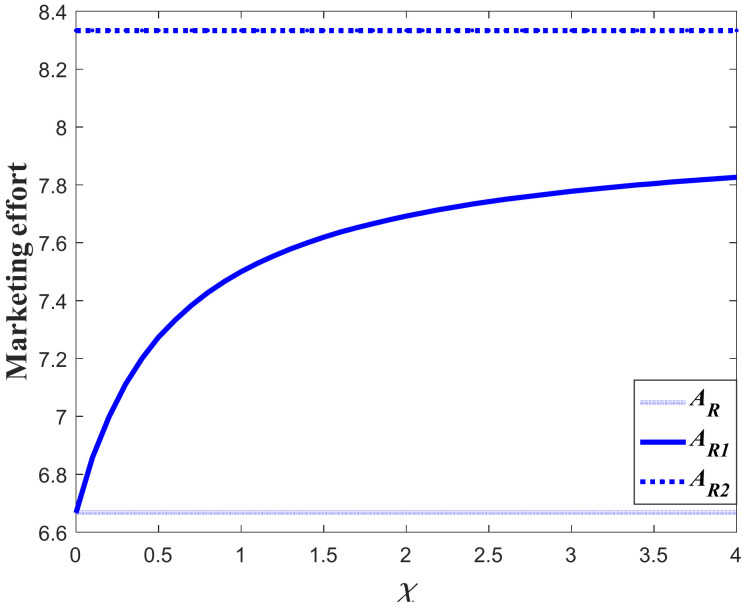
Marketing effort before and after the event (Scenario 2).

**Figure 15 ijerph-20-04644-f015:**
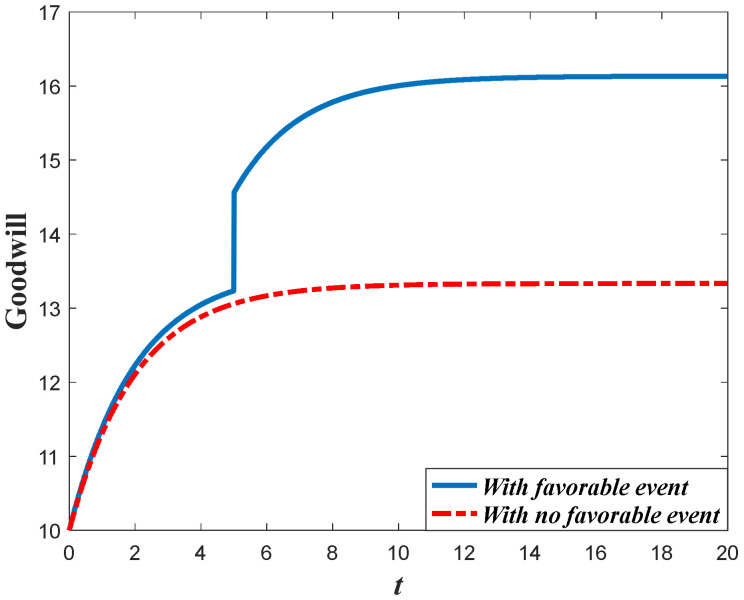
Goodwill level before and after the event (Scenario 2).

**Figure 16 ijerph-20-04644-f016:**
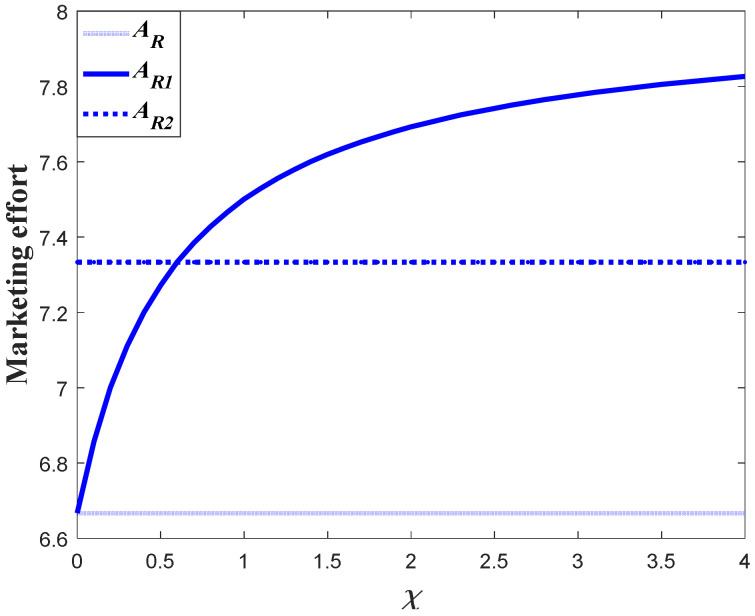
Marketing effort before and after the event (Scenario 3).

**Figure 17 ijerph-20-04644-f017:**
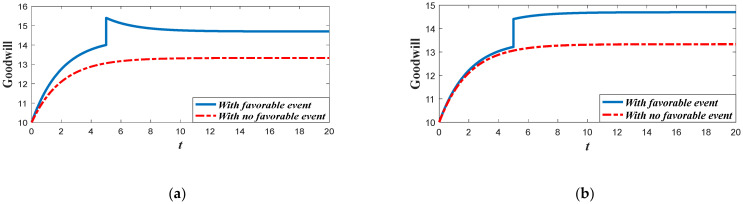
(**a**) Goodwill level before and after the event (Scenario 3). (**b**) Goodwill level before and after the event (Scenario 3).

**Table 1 ijerph-20-04644-t001:** Notations and definitions.

Notations	Definitions
Parameters	
$\gamma_{1}$,$\gamma_{2}$	Effectiveness of the marketing on goodwill before and after the event, $\gamma_{1},\gamma_{2}>0$
δ	Decay rate of the goodwill, δ>0
a	Baseline demand, a>0
θ	Effectiveness of the goodwill level on demand, θ>0
λ	Effectiveness of the carbon reduction effort, λ>0
ϕ	Unit carbon emission, ϕ>0
$\phi_{B}$	Baseline unit carbon emission, $0<\phi_{B}\leq \phi$
$p_{0}$	Price of carbon emission quota, $p_{0}>0$
$\rho_{M}$	The manufacturer’s marginal profit, $\rho_{M}>0$
$\rho_{R}$	The retailer’s marginal profit, $\rho_{R}>0$
$k_{M}$	The manufacturer’s cost coefficient, $k_{M}>0$
$k_{R}$	The retailer’s cost coefficient, $k_{R}>0$
χ	Occurrence rate of the favorable event, χ>0
ε	Expansion rate of the goodwill due to the favorable event, ε>0
$J_{i1}$ $,J_{i2}$	Net revenues of the manufacturer and the retailer before and after the event
Variables	
$I_{M}\left(t\right)$	Carbon reduction effort decision at time t
$A_{R}\left(t\right)$	Marketing effort decision at time t
E(t)	Emission level at time t
G(t)	Goodwill level at time t

## Data Availability

Not applicable.

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
