# Peer review of "Study of Carbon Reduction and Marketing Decisions with the Envisioning of a Favorable Event under Cap-and-Trade Regulation"

_ijerph, 2023, doi:10.3390/ijerph20054644_

Round 1

Reviewer 1 Report

First of all, I would like to thank the editor for giving me this opportunity to review the manuscript, and I would also like to congratulate the authors for the work they have done, as it shows great thoroughness and professionalism.

I also congratulate the authors because throughout the reading and analysis of the work one can appreciate the quality of the same as well as the ease of reading and understanding as it uses easily understandable language.

Likewise, the idea defended and the objective pursued in the research work provides a new step forward in the reduction of carbon and marketing strategies designed to establish maximum limits on these carbon emissions, which is related to the objectives pursued and the 2030 development agenda. I encourage the authors to continue their research in this area.

However, I would advise that some improvements have been taken into account, among which the following stand out:

Firstly, the percentage of plagiarism detected is 23%, which can be considered high and I advise authors to reduce it if their aim is to publish their manuscript in a high impact journal. One of the main requirements of any publication should be a reduced percentage of plagiarism, which denotes the authorship of the work.

In the part of the abstract it might be advisable to justify why the selected methodology has been used, without forgetting that this section should be reduced, so a simple allusion in one sentence would be sufficient.

It hurts in the subscription the authors because within the introduction section the research objective is summarised as the research gas covered as well as the main contributions that contribute to the research carried out without forgetting a brief reference to the parts that the research will contain.

Regarding the literature review section, it is clearly exposed by the different parts that can be seen within the pursued objective such as regulations, sustainability, marketing and control problems.

Perhaps it would be appropriate to focus on the research area covered by the research in each of the sub-sections.

Regarding sections 3 and 4, congratulations to the authors because the objective of the section is clearly specified and the model to be used is summarised. Perhaps within this section it would be convenient to allude to some type of research that has used the same method of analysis and makes reference to the degree of research covered with ours.

Similarly, in section six, it would be useful to support all the analysis with a bibliographical citation that defends the analysis carried out.

Section seven clearly describes the numerical result achieved in each of the hypotheses initially put forward. However, there is no section in the research work in which a discussion is made between the results achieved and the existing literature on the subject to date, unless a section is included in which a brief discussion of the results is made and the manuscript is not limited to describing the results achieved, all of which is supported by the corresponding bibliographical citations.

Regarding the section on the conclusion of the authors' subscription, because it clearly summarises the research objective, the research age covered and the main results achieved, however, a brief allusion to the limitations of the work and the authors' future line of research is missing.

Once all these improvements have been analysed, it would be necessary to carry out a new review before making a decision on publication.

Author Response

Dear Reviewer:

Thank you so much for your extensive reading and review of this paper. We all feel appreciated for the previous suggestions and comments which are very constructive to the betterment of this research. We have tried our best to revise this paper according to your comments and we hope the revisions is appropriate and satisfactory. The comments and corresponding replays are as follows:

Comment 1: Firstly, the percentage of plagiarism detected is 23%, which can be considered high and I advise authors to reduce it if their aim is to publish their manuscript in a high impact journal. One of the main requirements of any publication should be a reduced percentage of plagiarism, which denotes the authorship of the work.

Reply to comment 1: Thank you so much for point out this critical issue. We have tried our best to revise the paper according to the report you sent and we hope this issue could be improved.

Comment 2: “In the part of the abstract it might be advisable to justify why the selected methodology has been used, without forgetting that this section should be reduced, so a simple allusion in one sentence would be sufficient.”

Reply to comment 2: Thank you so much for this advice! Illustrating the methodology is essential part to the study. Hence, we briefly introduce the methodology adopted in this paper in the abstract part. We emphasize that “Since the event occurs randomly during the planning period, we use the Markov random process to depict the event and use differential game methodology to dynamically study this issue.”

Comment 3: “It hurts in the subscription the authors because within the introduction section the research objective is summarized as the research gas covered as well as the main contributions that contribute to the research carried out without forgetting a brief reference to the parts that the research will contain.”

Reply to comment 3: Thank you so much for raising this suggestion, we have tried to better the contribution part according to your guide in the revision version. In this part, we further briefly refer to the parts that the research will contain. I hope the revision is satisfactory to your suggestion!

Comment 4: “Perhaps it would be appropriate to focus on the research area covered by the research in each of the sub-sections.”

Reply to comment 4: Thank you so much for this advice! We have streamlined the literatures in each sub-section to make it more focused. In the literatures concerning the study of cap-and-trade, we concentrate on the stream mainly concerning the carbon reduction technology investment decisions under the cap-and-trade regulation, which is most relevant to this study. Another stream of literatures related concerns the study of sustainable supply chain by using differential game and we mainly focus on the researches using differential game model to study the carbon reduction strategy in sustainable supply chain, based on which we further study the carbon reduction strategy with consideration of favorable event. We also streamline the literatures concerning the control problem by deleting the literatures using Wiener process to tackle the uncertainty and keep those researches with Markov process, which constitutes major foundation of this paper.

Comment 5: “Regarding sections 3 and 4, congratulations to the authors because the objective of the section is clearly specified and the model to be used is summarized. Perhaps within this section it would be convenient to allude to some type of research that has used the same method of analysis and makes reference to the degree of research covered with ours.”

Reply to comment 5: Thank you so much for pointing out this insufficiency! We try to make up this point in the revision version. Concerning the literatures related to model construction in this paper, we refer to two streams of literatures. First stream underlies the construction the cap-and-trade regulation in this paper. The other stream underlies the construction of the favorable event. In the first stream, we refer to the following researches concerning the modelling of cap-and-trade regulation.

  1. Wei, C.; Zhang, L.F.; Du, H.Y. Impact of cap-and-trade mechanisms on investments in renewable energy and marketing effort. Prod. Consump. 2021, 28, 1333–1342.
  2. Ji, J.; Zhang, Z.; Yang, L. Comparisons of Initial Carbon Allowance Allocation Rules in an O2O Retail Supply Chain with the Cap-and-Trade Regulation. J. Prod. Econ. 2017, 187, 68–84.
  3. Chen, W.; Chen, J.; Ma, Y. Renewable energy investment and carbon emissions under cap-and-trade mechanisms. Clean. Prod. 2021, 278, 123341.
  4. Xu, L.; Wang, C.; Zhao, J. Decision and coordination in the dual-channel supply chain considering cap-and-trade regulation. Clean. Prod. 2018, 197: 551-561.
  5. Cao, K.; Xu, X; Wu, Q. et al. Optimal production and carbon emission reduction level under cap-and-trade and low carbon subsidy policies. Clean. Prod. 2017, 167: 505-513.

We involve literature [1-3] as the reference to the benchmarking rule which is adopted in this paper when deciding the quota of in the cap-and-trade regulation. According to the benchmarking rule, the unit carbon quota is allocated to the company. Meanwhile, we involve literatures [4-5] as reference to the construction of the carbon abatement decision, in which the carbon abatement effort is used to lower the unit carbon emission. Hence, based on the literatures [1-5], we further set up a differential game model with consideration of cap-and-trade regulation and the carbon abatement effort. We illustrate this point in the revision version.

In the second stream, we refer to the following researches to formulate the model

  1. Lu, L.; Navas, J. Advertising and quality improving strategies in a supply chain when facing potential crises. J. Oper. Res, 2021, 288(3): 839-851.
  2. Wang, W.; Hu, J. Optimal strategies of retailers facing potential crisis in an online-to-offline supply Chain. Probl. Eng. 2021, 2021: 1-18.

Enlightened by the researches concerning the study of brand crisis [6-7], we also use the Markov process to depict the favorable event which also randomly occurs but will elevate the goodwill level. Literatures [1-7] underlie the construction of the model in this paper. They use the similar method to conduct their analyses. However, we make some advances based on the above researches. For example, literatures [1-5] study the cap-and-trade regulation by using static optimization theory, based on which we continue the study of cap-and-trade regulation by using dynamic optimization theory. Meanwhile, based on literatures [6-7], we further study the favorable event under the cap-trade-regulation.

Comment 6: “Similarly, in section six, it would be useful to support all the analysis with a bibliographical citation that defends the analysis carried out.”

Reply to comment 6: Thanks so much for raising this point. Extant literatures have studied the brand crisis by introducing the Markov process and discussed the impacts of the crisis on marketing and operational strategies. Such as

  1. Lu, L.; Navas, J. Advertising and quality improving strategies in a supply chain when facing potential crises. J. Oper. Res, 2021, 288(3): 839-851.
  2. Wang, W.; Hu, J. Optimal strategies of retailers facing potential crisis in an online-to-offline supply Chain. Probl. Eng. 2021, 2021: 1-18.
  3. Wang, W.; Ma, D.; Hu, J. Optimal Dynamic Advertising Strategies with Presence of Probable Product-Harm Chinese J. Manage. Sci. 2022, 30(2): 204-216.
  4. Zhang, L.; Ma, D.; Hu, J. Research on the sustainable operation of low-carbon tourism supply chain under sudden crisis prediction. Sustainability, 2021, 13(15): 8228.
  5. Rubel, O.; Naik, P. A. Srinivasan S. Optimal advertising when envisioning a product-harm crisis. Sci. 2011, 30(6): 1048-1065.

However, as far as we know, extant researches rarely use the Markov process to study the favorable event. Similar to the crisis event, any company may have real chance to encounter a favorable event during the operation process but contrary to the brand crisis which exerts damage on goodwill level, the favorable event will elevate the goodwill and therefore increase the market demand. Hence, we use the Markov process to study this event. Meanwhile, we further study the favorable event and its impacts on the carbon abatement and marketing decisions under the cap-and-trade regulation, due to the fact that the elevation of goodwill tends to be associated with the increase of the market demand and carbon emission. Since not many literatures have discussed this issue before, we may have drawn some new conclusions. Even so, our findings are connected to the results in the researches mentioned above. For example, in literature [1-3], they found that the brand crisis will decrease the marketing effort before and after the event. However, when the players come into a favorable event as illustrated in this paper, they should increase the marketing effort before and after the favorable event.

Comment 7: “Section seven clearly describes the numerical result achieved in each of the hypotheses initially put forward. However, there is no section in the research work in which a discussion is made between the results achieved and the existing literature on the subject to date, unless a section is included in which a brief discussion of the results is made and the manuscript is not limited to describing the results achieved, all of which is supported by the corresponding bibliographical citations.”

Reply to comment 8: Thank you so much for pointing out this instructive guide! We compare the results in this paper to those in the extant literatures in the revision version. It is interesting to put the different results together and make comparisons. For example, we can find that in the literatures about the study of brand, the best decision of supply chain members is to decrease the marketing investment before the crisis. However, if the members may encounter the favorable event, which will elevate the goodwill, then the optimal decision turn out to elevate the marketing investment before the event. Hence, what strategy the company should adopt depend on what kind of event the company may encounter. We also compare the carbon reduction decision in this paper to those studies with no consideration of favorable event.

Comment 8: “Regarding the section on the conclusion of the authors' subscription, because it clearly summarizes the research objective, the research age covered and the main results achieved, however, a brief allusion to the limitations of the work and the authors' future line of research is missing.”

Reply to comment 8: Thank you so much for pointing out this insufficiency! We try to make up this point by adding a section elaborating on the limitations and future suggestions after the conclusion section. The limitations and future suggestions can be concluded as: 1) we only study the benchmarking rule in the cap-trade-regulation. In the future, we can study and compare different mechanism of the cap-trade-regulation with consideration of the favorable event; 2) we fail to incorporate the pricing problem in this research. In the future, we can further study the impacts of favorable event and the cap-and-trade regulation on the pricing of supply chain members to see how the members to adjust their pricing strategies to optimize their profits; 3) we assume that the carbon trading price and carbon quota to be exogenous variables instead of control variables. In the future, if we involve the policymakers as a member of the game, we may discover how the policymakers make decisions on carbon trade price and carbon quota from perspective of achieving optimal social welfare.

Finally, thank you so much again for your extensive review of this paper and we all feel appreciated for your constructive advice, which is of great importance to the betterment of this paper. We hope that our revisions and replies are appropriate and satisfactory! Thank you very much!

Best Regard!

All authors

Reviewer 2 Report

This paper address highlights the used of Markov process to re-evaluate the carbon reduction and marketing strategies. My suggestion as follows.

1. Need to have more detail issues highlighted with recent citation. Please link to any SDG goals

2. Is there any diagnostic or robustness analysis conducted to ensure the validity of the outcomes?

3. How the findings of this studies help the governemnt policymakers in solving the issues?

4. Who validate the formulation proposed in this study?

Author Response

Dear reviewer:

Thanks so much for your comments and suggestions for this research, which enlightens us of the insufficiencies and help us to improve this paper. Hence, these comments are very valuable and constructive. We have studied these comments carefully and made replies to the comments in the following section. We hope our replies are satisfactory and is capable of answering your question.

Comment 1: “Need to have more detail issues highlighted with recent citation. Please link to any SDG goals.”

Reply to comment 1: Thank you so much for inspiring us of this point! The SDGs such as Goal 12 (Responsible Consumption and Production) and Goal 13 (Climate Action) are closely related to the theme of this paper. Hence, we take the SDGs as the background in the revision of this paper and citate literatures relevant to make the paper more compelling.

Comment 2: “Is there any diagnostic or robustness analysis conducted to ensure the validity of the outcomes?”

Reply to comment 2: Thanks so much for raising questions about the validity of this paper. Extant literatures have studied the brand crisis by introducing the Markov process and discussed the impacts of the crisis on marketing and operational strategies. Such as

  1. Lu, L.; Navas, J. Advertising and quality improving strategies in a supply chain when facing potential crises. J. Oper. Res, 2021, 288(3): 839-851.
  2. Wang, W.; Hu, J. Optimal strategies of retailers facing potential crisis in an online-to-offline supply Chain. Probl. Eng. 2021, 2021: 1-18.
  3. Wang, W.; Ma, D.; Hu, J. Optimal Dynamic Advertising Strategies with Presence of Probable Product-Harm Chinese J. Manage. Sci. 2022, 30(2): 204-216.
  4. Zhang, L.; Ma, D.; Hu, J. Research on the sustainable operation of low-carbon tourism supply chain under sudden crisis prediction. Sustainability, 2021, 13(15): 8228.
  5. Rubel, O.; Naik, P. A. Srinivasan S. Optimal advertising when envisioning a product-harm crisis. Sci. 2011, 30(6): 1048-1065.

However, as far as we know, extant researches rarely use the Markov process to study the favorable event. Similar to the crisis event, any company may have real chance to encounter a favorable event during the operation process but contrary to the brand crisis which exerts damage on goodwill level, the favorable event will elevate the goodwill and therefore increase the market demand. Hence, we use the Markov process to study this event. Meanwhile, we further study the favorable event and its impacts on the carbon abatement and marketing decisions under the cap-and-trade regulation, due to the fact that the elevation of goodwill tends to be associated with the increase of the market demand and carbon emission. Since not many literatures have discussed this issue before, we may have drawn some new conclusions. Even so, our findings are connected to the results in the researches mentioned above. For example, in literature [1-3], they found that the brand crisis will decrease the marketing effort before and after the event. However, when the players come into a favorable event as illustrated in this paper, they should increase the marketing effort before and after the favorable event.

Comment 3: “How the findings of this studies help the government policymakers in solving the issues?”

Reply to comment 3: Thank you so much for raising this suggestion which is very constructive to the improvement of this paper. According to the results of this paper, the favorable event may help to lower the carbon emission when the unit carbon emission is relatively low and the carbon emission further decreases with the increase of expansion rate of the goodwill level. Hence, if the government aims to lower the carbon emission, he may manipulate the event to achieve this purpose. For example, when a company face such event and the unit carbon emission is relatively low, the government can help to advertise the event so that the expansion rate will further increase and the carbon emission will be further decreased. But if the unit carbon emission is relatively large, it is undesirable for the government to advertise the event from carbon reduction perspective. We have elaborated on this point from policymaker stand in the revision version. Thank you again for this excellent suggestion!

Comment 4: “Who validate the formulation proposed in this study?”

Reply to comment 4: Thank you so much for raising this problem! Two streams of literatures underlie the formulation of this study. First stream concerns the formulation of the cap-and-trade regulation. The other stream concerns the formulation of the favorable event. In the first stream, we refer to the following researches to formulate the model

  1. Wei, C.; Zhang, L.F.; Du, H.Y. Impact of cap-and-trade mechanisms on investments in renewable energy and marketing effort. Prod. Consump. 2021, 28, 1333–1342.
  2. Ji, J.; Zhang, Z.; Yang, L. Comparisons of Initial Carbon Allowance Allocation Rules in an O2O Retail Supply Chain with the Cap-and-Trade Regulation. J. Prod. Econ. 2017, 187, 68–84.
  3. Chen, W.; Chen, J.; Ma, Y. Renewable energy investment and carbon emissions under cap-and-trade mechanisms. Clean. Prod. 2021, 278, 123341.
  4. Xu, L.; Wang, C.; Zhao, J. Decision and coordination in the dual-channel supply chain considering cap-and-trade regulation. Clean. Prod. 2018, 197: 551-561.
  5. Cao, K.; Xu, X; Wu, Q. et al. Optimal production and carbon emission reduction level under cap-and-trade and low carbon subsidy policies. Clean. Prod. 2017, 167: 505-513.

We conform to the benchmarking rule when deciding the quota of in the cap-and-trade regulation as is shown in literature [1-3]. According to the benchmarking rule, the unit carbon quota is allocated to the company. Concerning the carbon abatement decision, we refer to literatures [4-5], in which the carbon abatement effort is used to lower the unit carbon emission. Hence, based on the literatures [1-5], we set up a model with consideration of cap-and-trade regulation and the carbon abatement effort. We illustrate this point in the revision version.

In the second stream, we refer to the following researches to formulate the model

  1. Lu, L.; Navas, J. Advertising and quality improving strategies in a supply chain when facing potential crises. J. Oper. Res, 2021, 288(3): 839-851.
  2. Wang, W.; Hu, J. Optimal strategies of retailers facing potential crisis in an online-to-offline supply Chain. Probl. Eng. 2021, 2021: 1-18.

The literatures [6-7] use the Markov process to depict the brand crisis event which exert damage on the goodwill level. Enlightened by their researches, we use the process to depict the favorable event which also randomly occurs but will elevate the goodwill level. Based on the literatures above, we formulate a differential game model with consideration of favorable event under the cap-and-trade regulation to explore the impacts of the event on the carbon abatement decisions

Finally, thank you so much again for your inspiring and constructive advice. You really enlighten us a lot about our study! We all feel appreciated for your effort and brilliant advices and we hope the answers we have offered are satisfactory!

Best Regard!

All authors

Round 2

Reviewer 1 Report

After reviewing the new version provided by the authors and having completed the manuscript with all the proposals, I consider that the manuscript can be published in its current form.

Reviewer 2 Report

The authors have done excellent job in revising their paper. Well done!